# Nanotechnology as a Novel Approach in Combating Microbes Providing an Alternative to Antibiotics

**DOI:** 10.3390/antibiotics10121473

**Published:** 2021-11-30

**Authors:** Bismillah Mubeen, Aunza Nayab Ansar, Rabia Rasool, Inam Ullah, Syed Sarim Imam, Sultan Alshehri, Mohammed M. Ghoneim, Sami I. Alzarea, Muhammad Shahid Nadeem, Imran Kazmi

**Affiliations:** 1Institute of Molecular Biology and Biotechnology, The University of Lahore, Lahore 54000, Pakistan; bismillah.mubeen@gmail.com (B.M.); aunzanayab01@gmail.com (A.N.A.); rabia.amjad545499@gmail.com (R.R.); inamgenetics@gmail.com (I.U.); 2Department of Pharmaceutics, College of Pharmacy, King Saud University, Riyadh 11451, Saudi Arabia; simam@ksu.edu.sa (S.S.I.); salshehri1@ksu.edu.sa (S.A.); 3Department of Pharmacy Practice, College of Pharmacy, AlMaarefa University, Ad Diriyah 13713, Saudi Arabia; mghoneim@mcst.edu.sa; 4Department of Pharmacology, College of Pharmacy, Jouf University, Sakaka 72341, Saudi Arabia; samisz@ju.edu.sa; 5Department of Biochemistry, Faculty of Science, King Abdulaziz University, Jeddah 21589, Saudi Arabia

**Keywords:** emergence of infectious disease, healthcare sector, multidrug resistance (MDR), antimicrobial resistance growth, nanotechnology-based innovation, pathogenic microorganisms, conventional antibiotics, biogenic nanoparticles

## Abstract

The emergence of infectious diseases promises to be one of the leading mortality factors in the healthcare sector. Although several drugs are available on the market, newly found microorganisms carrying multidrug resistance (MDR) against which existing drugs cannot function effectively, giving rise to escalated antibiotic dosage therapies and the need to develop novel drugs, which require time, money, and manpower. Thus, the exploitation of antimicrobials has led to the production of MDR bacteria, and their prevalence and growth are a major concern. Novel approaches to prevent antimicrobial drug resistance are in practice. Nanotechnology-based innovation provides physicians and patients the opportunity to overcome the crisis of drug resistance. Nanoparticles have promising potential in the healthcare sector. Recently, nanoparticles have been designed to address pathogenic microorganisms. A multitude of processes that can vary with various traits, including size, morphology, electrical charge, and surface coatings, allow researchers to develop novel composite antimicrobial substances for use in different applications performing antimicrobial activities. The antimicrobial activity of inorganic and carbon-based nanoparticles can be applied to various research, medical, and industrial uses in the future and offer a solution to the crisis of antimicrobial resistance to traditional approaches. Metal-based nanoparticles have also been extensively studied for many biomedical applications. In addition to reduced size and selectivity for bacteria, metal-based nanoparticles have proven effective against pathogens listed as a priority, according to the World Health Organization (WHO). Moreover, antimicrobial studies of nanoparticles were carried out not only in vitro but in vivo as well in order to investigate their efficacy. In addition, nanomaterials provide numerous opportunities for infection prevention, diagnosis, treatment, and biofilm control. This study emphasizes the antimicrobial effects of nanoparticles and contrasts nanoparticles’ with antibiotics’ role in the fight against pathogenic microorganisms. Future prospects revolve around developing new strategies and products to prevent, control, and treat microbial infections in humans and other animals, including viral infections seen in the current pandemic scenarios.

## 1. Introduction

Antibiotics are drugs that may either destroy microbes or suppress them. Based on their target population, antibiotics are graded as antibacterial, antifungal, and antiviral. The term antibiotic is generally used to refer to most antibacterial compounds [1]. Antibiotics have been used for generations to prevent and treat infections and aid with many medical therapies, varying from organ transplantation to chemical therapy. Inhibition of enzymes, DNA interference, RNA and protein synthesis, and destruction of the membrane structure are well-known antimicrobial pathways of antibiotics [2]. Depending on the formation and the mode of multi-drug resistance, many groups or classes of antibiotics have been developed. Thus, it is virtually impossible to imagine the future without antibiotics. 

Unfortunately, due to the rise of antibiotic resistance in microorganisms, this may become a reality [3]. Emerging drug resistance in microorganisms relates directly to the exploitation of antibiotics, excessive doses of medications leading to increased toxicity, prolonged hospitalization, and growing fatality [4], their widespread use in agriculture, and the lack of development of new antibiotics [5]. Moreover, the modern convenience of mobility (of products and infected people) helps disseminate pathogens across the globe at a rate never seen before [6]. Apart from the adverse socio-economic consequences, the public health challenge of antibiotic resistance to pandemic infectious diseases is serious [6]. World Health Organization (WHO) has recognized antimicrobial resistance (AMR) as one of the major challenges to public health [7]. 

Feasible pathways known to prevent antibiotic resistance in microbes include decreasing the consumption of antimicrobial medicines and improving drug release, modifying antibiotic targets, developing medicines for degrading or modifying enzymes in microorganisms, developing a biofilm coating containing the bacteria, and avoiding antibiotic exposure [8]. Eventually, such developments will lead towards reduced drug accumulation in microbial cells or a brief intracellular residency of drugs that do not effectively reach therapeutic concentrations [9]. Currently, however, higher doses and repeated drug administration are prevalent, contributing significantly to adverse side effects on animals and humans. Resistance has been developed towards several forms of antibiotics widely used against pathogenic microorganisms [10]. Most significantly, no new kinds of antibiotics have been developed in recent years. Furthermore, the development and promotion of new antibiotics is a costly and time-consuming procedure, requiring new compounds and multiple clinical studies and licensing [11]. The possibility that bacterial resistance towards all-new antibiotics will develop promptly would lead to decreasing antibiotic usage and declining sales, further exacerbating public health and economic condition. Therefore, the failure of antibiotic advancement would eventually lead to an elevated mortality threat from infections [12]. Consequentially, modern therapies are critical for overcoming these challenges.

Medicine has been modernized with the introduction of nanotechnology, the most significant breakthrough in recent years. There is a steady rise in the market for nanotechnology products. Nanotechnology, a groundbreaking science, will influence our efforts to improve human health. The medical industry has studied the longevity, efficiency, durability, flexibility, and inimitable physicochemical characteristics of nanoparticles. They are being utilized in numerous therapeutic approaches, such as the targeted delivery of medications, prognostic visual monitoring of therapy, and even tumor identification [13,14]. Several conventional approaches have been used to synthesize nanoparticles, for example, effective techniques such as physical vapor deposition, laser ablation, sputtering, melt mixing, and chemical methods such as photo-reduction, sol—gel, thermolysis, and micro-emulsion. As a result of these techniques, nanoparticles can become unstable, harmful compounds can attach to nanoparticles’ surface, and hazardous by-products can develop. 

The biogenic nanoparticle synthesis relies on green methodologies. Green synthesis advantages include the production of stable nanoparticles, the use of a biomass-based surface coating that provides extra active surface areas for biological interaction, the exclusion of dangerous formation of byproducts, and additional stabilizing or reducing factors that eventually make the procedure economical [15,16].However, constant human exposure to nanoparticles in the work environment may lead to unexpected danger to human health. Furthermore, secondary exposure to nanoparticles may occur through inhaling nanoparticles in the form of atmospheric toxins. Occasionally, these inhaled nanoparticles escape the immune system and are dispersed throughout the body, creating issues with systemic health.

In this study, we concentrated on antimicrobial drug resistance, nanoparticles, and their correlation with conventional antibiotics.

## 2. Microbes

Microbes are single-celled micro-organisms. They are so small that thousands of them can fit onto the point of a needle and can only be seen through a microscope. Microbes are the oldest forms of life on our planet [17], and microbial traces date back to over 3.5 billion years ago. The waste would not rot without microbial growth, and therefore less air would be accessible to breathe. Such microbes are present nearly everywhere–in food, soil, bone, water, humans, animals, and plants. Microbes are also considered microscopic creatures because they can exist everywhere around the environment. Some of them thrive in the cold, and others survive the extreme heat. Certain microbes require oxygen while others do not. Some microbes primarily induce certain forms of infectious conditions [17,18]. Fungi, parasites, bacteria, and viruses are all different classes of microbes. There are several common infections that are triggered by microbes. For example, Influenza can be caused by bacteria, fungi, protozoa, or viruses [19]. Inflammatory Bowel Disease is caused by bacteria [20]. Onychomycosis is caused by fungi [21]. Severe Acute Respiratory Syndrome (SARS) is caused by a virus [22]. Babesiosis is caused by Protozoa [23]. Protothecosis is caused by algae [24]. Gastrointestinal endogenous is caused by Archea [25,26] (Figure 1).

### Infectious Microbial Species

The development and effects of a disease on the human body depend on the form and strain of the pathogen [27]. The immune system generally provides an excellent defense against infectious agents. However, microbes may overtake the capacity of the immune system to fend them off. In such cases, an infection is harmful. Some microbes have no negative effect, while others generate toxins or toxic agents that cause adverse body responses. Due to this difference, some infections are harmless and hardly detectable, while others may be severe and life-threatening. Infection can spread in various ways, partly due to the differences in microbes themselves; fungi, parasites, bacteria, and viruses are all different varieties of microbes. These differences result from the mechanism of action on the body and the genetic content, function, shape, and size of the microbe.

## 3. Conventional Antibiotics

With a historical context of human disease, an exceedingly high percentage of infectious agents has been involved in infectious diseases. Microbes have been considered responsible for many infectious diseases during the second half of the nineteenth century. As a result, antimicrobial chemotherapy was introduced as the principal therapeutic technique against the pathogenic species.

Penicillin was discovered in 1928 by Fleming. In a region containing an infected blue mold (a fungus of the *penicillium* genus), the development of *Staphylococcus aureus* in cultivation dishes was inhibited, leading to the discovery that a microbe is developing compounds that may prevent other microbial growth. Subsequently, the antibiotic penicillin came into clinical usage in the 1940s. In the period of antimicrobial chemotherapy, penicillin was an excellent safety and efficacy agent, and during the Second World War, it saved the lives of many wounded troops. Penicillin had become the first antibiotic and marked the beginning of modern antibiotic production [5] (Table 1).

### 3.1. Mode of Action of Antibiotics

Antibiotics can be classified according to their mode of action, the spectrum of the action, or their chemical structure. Bactericidal or bacteriostatic antibiotics may be commonly present across the Gram-negative and Gram-positive spectrum [156]. Depending on molecular structure, they can be classed as macrolides, β lactam, aminoglycosides, glycopeptides, tetracycline, and quinolones. The bacteria are either bacteriological or bacteriostatic; the targets can be of a wide variety (Gram-negatives or Gram-positive bacteria) [5].

β-Lactam inhibits the growth of bacterial cell walls by binding penicillin-binding enzymes (PBPs). PBPs perform the function of connecting the peptide units in the peptidoglycan sheet. After β- lactams are connected to PBPs, cell lysis occurs. The lactam antibiotics are also divided into monobactams, cephalosporins, carbapenems, carbacephems, and penicillins. Penicillin-resistant bacteria were reported to appear in the late 1960s. The enzymes β-lactamases, which could degrade β-lactam antibiotics, were synthesized by these bacteria. However, the emergence of carbapenem, the new class of β-lactams, solved this problem as carbapenem is not sensitive to the β-lactamases. Carbapenems exhibit the broadest spectrum of activity of all of the recognized lactams [5]. However, some bacterial species showed resistance to carbapenem [157].

Glycopeptides also attack the synthesis of bacterial cell walls, and additionally, they block the PBPs and inhibit peptidoglycan synthesis [5]. By attacking protein synthesis in the cell’s tetracycline, macrolides, oxazolidinones, and aminoglycosides block bacterial growth. The binding of macrolides to the 50S ribosomal subunit causes the inhibition of mRNA elongation during translation [5]. Oxazolidinones are also connected to the 50S subunit; however, they inhibit protein synthesis by inhibiting the development of a 70S initiation complex similar to macrolides [158]. These two groups combined make up the 50S category of blockers. Aminoglycosides and tetracycline attach to the ribosomal 30S subunit that prevents the utilization of aminoacyl-tRNA to the ribosome, thus inhibiting the synthesis of proteins. Tetracycline and macrolides are typically bacteriostatic, while the mode of action of aminoglycosides is broadly bactericidal [5]. 

Nucleic acid synthesis is necessary for cell survival. Quinolones inhibit bacterial growth by inhibiting the operation of helicases in DNA which, just before reproduction or repair of DNA, are crucial for relaxing DNA double-helical structure. Quinolones also interfere with bacterial functions of topoisomerase II and topoisomerase IV, which negatively impact RNA polymerase and thus inhibit the synthesis of RNA [5].

The para-aminobenzoic acid (PABA), a substrate for synthesizing folic acid used in bacterial cells, is imitated by sulfonamides structurally. Sulfonamide avoids cell division and causes the inhibition of the growth of bacteria, and it is necessary for the synthesis of nucleic acid. Regrettably, these modern antibiotics have been detected in bacteria which makes it difficult to treat infections due to these bacteria [159] (Figure 2).

### 3.2. Origin of Antibiotic Resistance

Antibiotic resistance is considered present when a drug begins to lose its ability to successfully suppress bacteria growth. In the presence of active antibiotics, bacteria are “immune” and continue to divide. Bacteria are called resistant bacteria when they replicate even in the presence of antibiotics [160]. If the microbes are less susceptible or resistant, an effect greater than the usual concentration of the same drug is needed. Antimicrobial resistance has been found immediately after the launch of new antimicrobial compounds [161]. The mechanism of natural selection, in which evolution allows all bacteria to have a degree of low resistance, may partly explain antibiotic resistance [162].For instance, one study confirmed that sulfamethoxazole and trimethoprim (TMP-SMZ), ampicillin, and tetracycline, commonly used in earlier years, now no longer play a role in Thailand’s treatment of non-cholera diarrhea [163]. However, a study in Bangladesh demonstrated the effective therapeutic use of the same drugs [164]. Even before the use of antibiotics in infection control, resistance has been documented [165].

Agricultural antibiotics are typically identical and adjacent to commonly used antibiotic compounds [166], which can also encourage drug resistance. The food chain may be viewed as the primary route of spreading antibiotic-resistant bacteria between the animal and human populations. Livestock obtains antibiotics from food, water, or parents who may bear microbial resistance to a specific antibiotic [166]. Antibiotic resistance in livestock feed increases with the use of antibiotics as growth promoters [167]. According to the study of the rural villages in Barcelona, a fecal carrier of Quinolone-resistant Escherichia coli was reported in one-fourth of the babies, the possible source of which could be poultry or swine. These children have been exposed to quinolones [168] (Table 2).

### 3.3. Development of Antibiotic Resistance

As suggested above, bacteria seem to have a natural mechanism that promotes resistance. The resistance mechanism arises by mutations at the gene stage [180]. Selective pressure is caused by antibiotics, and even the genes function in accordance with selective pressure [181]. Bacteria possess the ability to transfer genetic material directly between themselves by transferring plasmids, meaning that natural selection may not be the only process through which resistance develops. The bacteria in a colony thus may mutate, resulting in resistance [182]. Broad-spectrum antibiotic drugs used in the treatment of nosocomial infections in health centers may, in fact, improve microbial resistance since large colonies of mutated bacteria are often found in such places [183]. Increased association between antibiotic-resistant infections and antibiotic use has been demonstrated [184]. Resistance development can also arise in cases where patients fail to complete the course of their prescribed medication. In such cases, the bacteria remain unaffected and become much more immune to the action of antibiotics [181]. Thus, bacteria can acquire multiple resistance characteristics over time and become immune to many antibiotic classes [185,186,187]. Some FDA-approved antibiotics for the treatment of microbial infections and their resistant microbes are listed in Table 3.

The Antibiotic resistance in bacteria develops through the mechanisms shown in Figure 3:Antibiotic enzyme inactivation/degradation; an endogenous cellular enzyme is modified to interact with that of the antibiotic in a manner in which the bacteria are no longer affected. B-lactamase enzymes are among the most important examples; they hydrolyze most commonly administered antibiotics, i.e., b-lactams (cephalosporin and penicillin), and are the most widespread source of antibiotic resistance in Gram-negative bacteria.The excretion of the drug through efflux pumps; Bacteria are triggered to eliminate the antibiotic by stimulating the proteins that can eradicate an extensive range of substances from the periplasm to the outside cell. This is a mechanism of resistance especially essential for *P. aeruginosa* and *Acinetobacter spp*.Reduced absorption by variations in the external membrane permeability; these changes inhibit the successful entry of antibiotics.Drug target modifications to weaken or demolish the antibiotic binding efficacy and thereby minimize its potential.

### 3.4. Availability of Antibiotics

Antibiotic production is no longer deemed an economically viable strategy for pharmaceutical companies [11]. Because antibiotics are widely used and are primarily curative, they are less effective in managing conditions related to diabetes, psychological disease, asthma, or gastrointestinal complications [11,189].

An annual cost—benefit study by the Office of Health Economics in London reports that the current net value of the new antibiotic is $50 million compared with about 1 billion dollars in neuromuscular condition treatment drugs [11]. The expense of modern antibiotics is typically USD 1000–USD 3000 a year, while USD 100–USD 1000 is spent on cancer chemotherapy [11,190,191]. The effectiveness, usability, and generally lower prices of antibiotics have, on occasion, resulted in poor evaluation from investors and the general public [190].

Furthermore, restricted use of antibiotics has been recommended by microbiologists and infectious disease specialists [190]. In some cases, healthcare professionals limit the use of the latest medication, believing it might promote resistance, and instead, continue administering current antibiotics that have shown equivalent effectiveness prior to the introduction of a new antibiotic [190]. New drugs are therefore often used as the “last line” of defense medicine in the fight against severe diseases. Such a method limits the usage of modern drugs and reduces investor returns [190]. Many pharmaceutical companies fear that the million-dollar investments required to develop new antibiotics may provide inadequate returns [189,190,191]. Graph 1 shows a relative decrease in the manufacturing of new antibiotics over a period of time (Figure 4).

## 4. Nanomaterials

Nanotechnology-based approaches rang from the engineering and material sciences to biology and medicine. Materials with dimensions between 1–100 nm are commonly referred to as nanomaterial. They vary in shapes and sizes to help give characteristic features for a large spectrum of uses. The properties of the material undergo significant changes when limited to a rather small scale. Health sciences have shown considerable interest in various applications of nanotechnology. Nanomaterials are usually metal and metal oxides or their composite, carbon-based, and emulsion-based. They are cost-effective and can fight antibiotic-resistant bacteria. Approximately 10 million deaths by 2050 are predicted due to the growing risk of antibiotic resistance [192]. Nano-based technologies can offer a long-term and effective approach to managing drug resistance. 

### 4.1. History and Development of Nanomaterials

The protection of ceramic matrixes has been in use for over 4500 years, including the use of organic asbestos nanofibers [193]. More than 4000 years ago, the ancient Egyptians used NMs for synthesizing five nm diameter lead sulfide (PbS) nanoparticles (NPs) for hair dye [194]. NMs were based on the synthetic chemical process. “Egyptian blue” was the first synthetic stain produced and utilized by the ancient Egyptians in the third century BC with a synthetic combination in the range of nanometer-sized quartz and glass [195].

The synthesis by chemical methodologies of the metallic NPs dated back to the 14th century BC when Mesopotamians and Egyptians began the manufacture of glass from metals that could be referred to as the beginning of the metal nanoparticles era [196]. These materials may be the first experimental cases of synthetic NMs. In Frattesina di Rovigo (Italy), a red glass colored by Cu nanoparticles surface excitation was discovered during the late Bronze Age (1200–1000 BC) [197]. Likewise, it is recorded that Cu NPs and cuprous oxide (cuprite Cu_2_O) were found in Celtic red enamels from 400–100 BC [198]. However, the most notable example of prehistoric metallic NPs is a Roman glass workpiece [199]. The Mesopotamians began to use glazed ceramics for metallic decorations in the 9th century [193]. Moreover, clay minerals with a size of a few nanometers are the most ancient example of nanomaterial. In 5000 BC, in Cyprus, clay was used to bleach clothes and wool [200]. The synthesis of an AuNP colloidal solution, which was the very first scientific effort in NP preparation, was documented in 1857 by Michael Faraday. This was apparently among the first publications that detected and explored quantum-size effects. The cause of the particular colors of metal colloids was later explained by Mie in 1908 [201]. In the 1940s, SiO_2_ NPs were developed to provide rubber insulation to replace carbon black [202]. Today, manufactured nanomaterials can significantly enhance the properties of bulk materials with respect to their strength, conductivity, resilience, and lightness and can offer beneficial applications and serve as structural and detecting materials for protection. Despite other potential applications, taking advantage of the favorable shape and size to enhance the appearance of materials remains the primary use of NPs.

In 2003, Samsung launched the antibacterial technology Silver Nanomic^TM^ from ionic AgNPs in their air conditioners, washing machines, air purifiers, and cooling systems [203]. In auto production, NPs and nanostructured materials (NSMs) are commonly used as tire fillers to improve adherence to the surface, car body fillers to improve stability, and translucent layers for heating, mist, and ice-free window panes [204]. By the end of 2003, Mercedes-Benz launched an NP base coat for metallic and nonmetallic surfaces in series production. The coating improves the rubbing resistance and strengthens the gloss. The latest studies concentrated on the production of specialized earth-based astronomical telescopes with adaptive optics and magnetic mirrors with ferrofluid shape-shifting potential [205]. In solar cells with dye sensitization potential, TiO NPs are widely used [206]. In 2012, Logitech launched the first large commercial usage of dye-sensitized solar cells, an external light-powered iPad keypad. Abraxane^TM^ was developed in 2005, sold, and launched on the pharmaceutical market as a human serum-albumin NP substance comprising paclitaxel [207].

### 4.2. Nanoparticles Act as Antimicrobials

In the 1980s, the advance in nanotechnology allowed Nano-scale construction by the tracking of atomic particles. Subsequently, nanotechnology has become important in diverse areas like biomaterials, organic chemistry, medicine, and others. In the medical science and healthcare industry, nanomedicine and nano-scale particles [208] are used for therapeutic purposes, as biomaterials, and diagnostic tools [209]. Thus nanomaterials and future molecular nanotechnology are widespread medical applications [210,211]. Nanotechnology has made many difficult diagnoses possible and has expanded the knowledge of disease pathogenesis. Many medications and procedures are seriously impacted because of their poorer efficacy, usefulness, and adverse effects. As the scale of nanoparticles is close to biological molecules and forms, in vivo and in vitro therapeutic strategies and higher specificities are useful [208]. Higher specific activity medications can increase effectiveness and decrease adverse effects. Smaller-sized nanoparticles may have treatment application in the high-risk areas, reducing the potential harm and delivering the exact medication dosage required.

Despite their past therapeutic successes, the use of antibiotics became problematic at the beginning of the 20th century [212], with the development of drug-resistant bacteria brought on partly through their extensive use [213,214]. Increased pathogen antibiotic resistance to colistin [215], carbapenems [216,217], and tigecycline [218], and stalled production of new antibiotics made it impossible to cure the contagious disease caused by harmful pathogens. Even though a new antibiotic may be found, it could not provide a guarantee of its effectiveness against all multi-drug-resistant infections [219]. Nonetheless, the threat to public health is real since both the Gram-negative and Gram-positive multiple-drug-resistance bacteria are developing faster than ever [220]. Thus, technological developments must aid the efforts against these dangerous pathogens and help meet the long-term demand for successful treatment of drug-resistant bacterial infectious disease [221].

The latest developments in the production of medicinal nanomaterials may benefit the purpose of antibiotics. Theoretically, nanoparticles are a modern class of bacterial antimicrobials and synthetic pathogens. Nanoparticles as antibiotics can hopefully minimize resistance and support the efficient delivery of antibiotics [222]. Recent research suggests that certain antimicrobial-activated metal nano constructs have indeed helped combat infectious diseases [223]. Such constructs benefit from lower toxicity, lower cost, and better pharmacokinetic factors while helping to eliminate drug-resistant bacteria. Their most significant benefit is that they maintain their efficacy longer than traditional antibiotics, which is extremely valuable in the long-term sustained therapeutic effect [224].

### 4.3. Classification of Nanomaterials

The nanomaterials are classified into four categories:Carbon-based nanomaterialInorganic nanomaterialOrganic-based nanomaterialComposite-based nanomaterial

#### 4.3.1. Carbon-Based Nanomaterial

These nanomaterials typically have carbon and are found in spheres, ellipsoids, and hollow tubes. The types of carbon-based NMs include Fullerenes (C60), Carbon Nanotubes (CNT), Carbon nanofibers, Carbon Black, Graphene (Gr) [225] (Table 4).

##### Carbon Nanotubes (CNTs)

CNTs reported by 1991 are cylindrical informs connected with covalent bonds [226]. Multiwalled nanotubes and single-walled (SW) nanotubes have various types of single pipe CNTs 1–5 nm in diameter, and multiple nested tubes of lengths ranging from 100 nm to micrometers, respectively [227]. The cytotoxic influence of CNTs has been shown in both in vivo and alveolar macrophages [228,229]. The antimicrobial property of SWNTs indicates that their effective antibacterial and antiviral properties are based on the low aqueous dispersion of pure CNTs. The aqueous dispersion of CNTs has recently been demonstrated and could further be improved utilizing other surfactants or polymers. The best antimicrobial carbon-based nanomaterials identified are SWNTs. Initial interaction with bacteria, membrane dysfunction, and membrane oxidations are part of the SWNTs’ action in preventing microbial growth [230]. The use of CNTs for purifying water, *E. coli* inactivation, and poliovirus are being gradually studied, and MS2 phage elimination is also being investigated [231]. Thus far, CTTs offer relevant materials that can be used as antimicrobial agents.

##### Fullerenes

Fullerenes are ball clusters made up of atoms of carbon [232]. Antimicrobial activity was observed in fullerenes against many bacteria, including *Salmonella, Streptococcus spp,* and *E. coli* [232]. It Is suggested that their antibacterial activity was induced by energy metabolism inhibition after nanoparticles were internalized into the bacteria [233,234]. It was proposed that fullerene compounds prevent bacterial growth by destroying the respiratory chain [235].

The antibacterial efficacy of a sulfobutyl fullerene derivative in environmental bacteria was assessed by Yu et al. (2005). They observed that after photoirradiation, the used derivatives could inhibit environmental bacteria [236]. Mizuno et al. (2011) have also stated that cationic replacement fullerene derivatives are highly successful in destroying a large variety of microbial cells following the white light irradiation. A new generation of synthetic fullerenes was tested against Gram-negative and Gram-positive bacteria and whether they carry basic or quaternary amino groups.

##### Graphene Oxide (GO)

Graphene is generally referred to as a monolayer of carbon atoms, closely enclosed within a 2-dimensional crystal [237]. The GO Nanosheets are created by the chemical modification of graphene with suspended epoxyl, hydroxyl, and carboxyl groups that can be dispersed rapidly into water. The membrane stress is known to constitute the critical antimicrobial function of GO through direct interaction with sharp Nanosheets [238]. The inhibitory effects on *E. coli* growth were demonstrated by both graphene and GO. Testing antimicrobial effects on graphene sheets was performed by Akhavan and Ghaderi (2010), confirming that the direct relationship of extremely sharp edges and bacteria has resulted in RNA effluxes by impaired cellular membranes of both Gram-positive and Gram-negative bacterial membranes. The antimicrobial effects of the two active GO nanostructures (graphene oxide-chlorophyllin and graphene oxide-chlorophyllin-Zn) on *E. coli* were reported by Azimi et al. (2014). The authors suggested that *E. coli* was affected by the functional GO.

#### 4.3.2. Inorganic Nanomaterial

##### Silver Nanoparticles (AgNPs)

Silver has been used as an antibacterial agent for many years. Applications of silver as metal and metal oxide nanoparticles are included in nanomaterials. These nanomaterials can be synthesized into metals such as Ag or Au, metal oxides such as ZnO and TIO_2_ NP, and semiconductors such as ceramics and silicon. A few of these demonstrate high antimicrobial activity towards certain bacteria, viruses, and other microbes. Specific nanomaterials with strong antibacterial efficacy are thought to have high volume to surface ratios as well as a unique chemical—physical property [239] (Table 4).

Silver as an antibacterial agent has been used in various ways; as silver nitrate, silver sulfadiazine, and powdered silver for infectious diseases, dental research, catheters, and burned wound treatment [240]. The development of antibiotics in the 1940s limited the use of silver as an antibacterial agent. However, the emergence of antibiotic-resistant bacteria and diminished antibiotic efficaciousness has resurrected the therapeutic use of silver [241]. AgNPs are found suitable against almost all viruses, bacteria, and other eukaryotic microorganisms in various types of metal and metal oxides [242,243]. The effectiveness of AgNPs as antimicrobial agents depends entirely on their size and shape [244]. Mechanism of action of AgNP targets the cell division and respiratory chain, eventually destroying the cell [245]. Increased synergistic antimicrobial actions against Gram-Positive and Gram-Negative bacteria have been reported, resulting from the action of AgNPs in conjunction with conventional antibiotics such as vancomycin, erythromycin amoxicillin, and penicillin G [246].

Further studies suggested that Ag^+^ ion, which has a sulfur and nitrogen affinity, can suppress and destroy the protein structure by binding it to thiol and amino groups [247]. Lastly, silver NMs have been reported as photocatalytic [248] and capable of triggering reactive oxygen species ROS [249,250,251]. Some have denied that this effect is cell-type dependence, at least in eukaryotic cells [242]. Silver has a broad spectrum of applications, from injury dressing to surgical device coating and manufacturing applications. Often there are negative health consequences associated with the use of metallic silver [252]. These adverse effects include permanent skin pigmentation (argyria) and eye loss, organ destruction, inflammation, and changes in the blood cell count [253]. However, amounts of AgNPs commonly used, and the concentration-dependent toxicity that influences the mitochondrial activity (possibly responsible for AgNPs toxicity) is still below observation [254].

##### Gold Nanoparticles

Gold coating on carbon nanotubes improves drug production. Gold nanoparticles have been used to identify bacteria by adjusting colors to facilitate cytometry flow responsiveness. Gold nanoparticles’ action as antimicrobial agents is primarily based on the electrostatic appeal to the cell membrane’s negative-loaded bilayer; the suggestion has also been supported by the discovery that anionic particles are not toxic while cationic particles are [243]. Near-infrared (NIR) light can absorb gold nanocages, nanoparticles, nanorods, and nanoshells often used to cure a bacterial infection under intense laser beams of sufficient wavelength [252]. AuNPs, coupled with antibiotics or antimicrobials, were shown to have elevated effectiveness, and modifications have been studied of the specific antimicrobial effects [255]. In addition, bacterial killing was attributed to the intense laser-induced hyperthermic effect couped with the bubbles forming around cluster AuNPs [256]. The gold nanoparticles have been found to cause strong antimicrobial effects on Gram-negative and Gram-positive bacteria with an antibiotic (i.e., streptomycin-coated Au NPs, gentamicin, and neomycin). In contrast with conventional free ampicillin, nanotechnology advances show that AuNPs coated with chitosan and ampicillin demonstrate at least a two-fold rise in their antimicrobial activity [243]. Gold nanoparticles encourage adjuvants to replace antibiotic therapy to treat severe bacterial infections, including multi-drug-resistant bacteria, with low doses and minimal adverse effects. 

##### Zinc Oxide Nanoparticles

Zinc, as a metal oxide, an antibacterial agent ZnO NPs, and its antibacterial activity can conserve agricultural goods and food against specific foodborne bacterial pathogens such as *E. coli* [242,257]. ZnO NPs’ less expensive production, the ability to obstruct UV, and its white appearance make it more useful than Ag NPs. It showed the peerless capability of the multilayer avowal of Nano-ZnO on cotton fabric with antibacterial activity against *S. aureus* on its analysis part [258]. ZnO NPs’ potential to destroy the bacteria by destroying the cell membrane (proteins and lipids) and removing the intracellular material makes them suitable antibacterial particles. The ZnO-based nanoparticles are additionally broad-spectrum bactericidal NM [259] and have shown a wide variety of antibiotic activity against different microbes that mainly depend on the particle size and concentration chosen [260]. In addition, it also generates H_2_O_2_ and Zn^+2^ ions vital to bacterial cells [260]. Polyvinyl alcohol (PVA)-veiled ZnO shows enhanced permeability of the membrane, cell idealization, and intracellular structural adjustments [261].

##### Titanium Dioxide Nanoparticles

Titanium Dioxide (TiO_2)_ is a nontoxic antimicrobial with a potential bactericidal application. It is most frequently employed as a photocatalyst disinfecting material. Titanium dioxide nanoparticles (TiO_2_ NPs) have been widely studied and compared with other preferred nanoparticles to assess their antimicrobial photocatalyst activity [262]. The inhibition of microbial growth is found to be higher after irradiation with near-UV light, and the bactericidal behavior of TiO_2_ NP improves noticeably when combined with UV-A [263]. TiO_2_ behavior largely depends on the scale, intensity, and wavelength of light (100–1000 ppm levels are required to destroy bacteria). The UV-A irradiated TiO_2_ antibacterial agent has shown reduction efficacy in microbial growth (in the declining order) of *P. aeruginosa, E. coli, S. aureus, C. albicans,* and *E. faecium* based Cell membrane [264]. It was also suggested that the antibacterial photocatalytic efficacy of TiO_2_ conditional density of the microbial surface morphology starts to fall in the sequence of viruses > bacterial wall > bacterial spores [264]. Growth inhibition characteristics of TiO_2_ NPs with UV-A irradiation show inhibition against *Enterobacter cloacae*, *E. coli*, and *P. aeruginosa* were surprisingly less successful against *Enterobacter cloacae* than *P. aeruginosa* and *E. coli* [265]. 

Reactive oxygen species (ROS) production, free hydroxyl radicals, and peroxides render TiO_2_ a useful antibacterial photocatalytic agent [266]. TiO_2_ NP oxidative strike has been attained with hydroxyl radicals (potent oxidants) created from TiO_2_ photocatalyst, which have broad (nonselective) reactivity and mainly attack microbial surfaces. The composition and the integrity of the cell wall are essential to sustain semi-permeability, respiration, and other phosphorylation reaction. TiO_2_ NPs photocatalytic action can interrupt the normal cell function compromising the bacterial cell membrane structure. The TiO_2_ photocatalyst action leads to lipid peroxidation reaction (oxidative lipid degradation), which subsequently leads to cell death [267]. Thus, while the usage of radiation improves the photocatalytic antibacterial activity of TiO_2_ NPs, it also causes individual bacterial mortality from irradiation [268]. The potential for visible light activation (e.g., sunlight) makes TiO_2_ far more remarkable antibacterial agent possible. 

Metal doping improves the TiO_2_ antibacterial properties, leading to enhanced bacterial and viral photocatalytic inactivation [269]. Composite Ag/(C, S)-TiO_2_ NPs have shown strong light-independent antimicrobial activity when tested against *B. subtilis* and *E. coli* spores [270]. TiO_2_ can also be used as a low-cost, high-efficiency, stable, nontoxic alternative to traditional chemical disinfectants, which may produce toxic byproducts, and as such, it would be of particular value for water treatment systems in developing countries. [227]. TiO_2_ reactivity toward microorganisms can be applied to improve food safety, hygiene, and cosmetics using TiO_2_ photocatalytic disinfection of nanocomposite antimicrobial surface coating capable of destroying UV radiation-tolerant microbes [269]. TiO_2_ nanocomposite surface coating for orthodontic products, toothbrushes, dental implants, and screws has shown effective antibacterial activity against *Lactobacillus acidophilus* [266].

##### Copper Nanoparticles

Copper is a beneficial antibacterial agent since it is a structural part of many microorganism enzymes. When Cu^2+^ ions are high, ROS can be formed, disrupting amino acid formation and a more toxic DNA [271,272]. The basic principle behind copper microbial inactivation is the so-called contact-killing. Other antimicrobial mechanisms such as inoculation methods and incubation period, which are important, also rely on the copper contact method. The efficiency of this touch extinction has been improved by factors such as elevated temperatures [273], high metal content in copper [274], and decreased relative moisture [275]. Cu NPs with great affinity to the amines and carboxyl group on the surface of the organism kill *Bacillus subtilis* and have demonstrated better action than Ag NPs [276]. The chemical and physical quality of copper oxide (CuO) compared with silver is less expansive [277].

##### Aluminum Oxide Nanoparticles

The bacterial cell wall was damaged at a higher concentration of aluminum oxide (Al_2_O_3_) nanoparticles [278]. Alumina NPs are high-temperature thermodynamically stable particles. The chemical arrangement includes oxygen atoms that fill a matrix to two-thirds of the octahedral sites [278], which conform to hexagonally wrapped alumina ions. The aluminum oxide NPs had an effect on the surface charge, shape, and particle size [279]. In contact with organic matter, nanoparticles appear to mix with hard water and seawater in soil. Such aggregations rely on the pH and salt content of the particles. Most toxicity studies focus on the detailed configuration and characterization of the solution parameters, including thickness, distribution, structure, morphologies, surface areas, surface chemistry, and particle reactivity. They are believed to have moderate inhibitory properties and may be used in combination. Ag/Meso-Al_2_O_3_ nanoparticles displayed extensive *S. aureus* and *P. aeruginosa* inhibitory activity.

##### Nitric Oxide (NO)—Releasing Nanoparticles

Nitric oxide (NO) has an immune function as a diatomic free radical and is an effective antimicrobial agent acting against infection in two ways; at a low concentration at which it promotes the growth of immune cells, and at a high concentration where it inhibits or kills pathogens (via binding with DAN, proteins, and lipids). For combination therapy, NO is commonly used, but the performance of individual NO donors, based on sparse evaluation, shows limited results. The difficulty of processing or administering NO as an antibacterial agent has been a restricting factor. Lately, a strong trend of vulnerability has been found in Gram-negative and Gram-positive bacteria, including methicillin-resistant *Staphylococcus aureus,* by utilizing a gaseous NO (gNO) donor delivery platform [280]. 

NP-based scaffolding could hold large NO loads, which can be released at appropriate sites, physiological pH, or temperature under aquatic conditions [281]. The NO and NO-releasing Silica Nanoparticle antibacterial experiments have been conducted to destroy Gram-positive and Gram-negative fungi and bacteria. In vitro MRSA study has shown pathogen behavior in silane with nanoparticles releasing NO [280]. These nanoparticles can associate with the pathogen surface causing pathogen inhibition and leading to bactericide. The NO-releasing NPs are versatile with unique properties dependent on nanoparticles’ size and adjustable NO donor design. NO-releasing NPs have been used for treating contaminated wounds and subsequently for successful healing of wounds in diabetic mice [282]. Human studies have been limited thus far, but given the versatility and potential of NPs that make them highly suitable for NO delivery, there is a need for intensified research into the human application of this technology.

##### Magnesium Oxide Nanoparticles

Magnesium oxide (MgO-NPs) is an essential mechanism behind antimicrobial activity; it produces ROS and other nanoparticles [283,284]. MgO-NPs are associated physically with the cell surface as some other nanoparticles and cause a dysfunction of the cell membrane integrity which contributes to membrane leakage [8]. Therefore, MgO-NPs destroy the cells through intracellular biomolecules irreversible oxidation. Furthermore, some studies have shown that with the lack of lipid peroxidation and ROS, MgO-NPs generate a high antibacterial effect. Research suggests that the relationship between MgO-NPs antibacterial behavior and nanoparticle association to the microbial cellular membrane changes pH and releases Mg^+2^ [285]. Moreover, the antibiotic behavior of MgO-NPs was found to be caused by the adsorption of halogen molecules onto the exterior of MgO [8].

##### Iron Oxide Nanoparticles

Iron oxide nanoparticles are highly ferromagnetic and have reduced oxidation sensitivity. There has been considerable attention towards iron oxide nanoparticles since they are known as nontoxic and biologically compatible materials because of the presence of Fe (II/III) ions [286]. Release and termination of toxic ions, oxidation damage caused by catalysis, the ion cell membrane transport activity variation, and lipid peroxidation or surfactant products are chemical pathways associated with reactive oxygen species (ROS). In nanotoxicology, ROS is considered a key rudimentary chemical mechanism, which can result in secondary processes that can eventually destroy cells and even cause cell death. In addition, ROS are a major inflammatory factor. The inflammatory action is believed to occur by up-regulation, stimulated by the activation of specific transcription factors in genes involved in a proinflammatory response (NF-ŢB, AP-2). Cell stability can also be directly affected by free radical formation [287,288]. Nano–bio-interface physical processes are primarily governed by particle size and surface properties. They involve membrane function, membrane transport mechanism destruction, protein conformation or folding, and the accumulation of proteins [289]. The iron oxide nanoparticles can impede the production of *B. subtilis* as well as *E. coli*. The maximum inhibition (29 mm) in *S. aureus* compared to *E. coli* and *P. aeruginosa* was observed with 015 mg/mL of iron oxide nanoparticles [290]. In another analysis, Fe_2_O_3_ and Ag/Fe_2_O_3_ were found to have antibacterial properties, and nanoparticles against *S. dysenteriae* had antibacterial activity much more significant than that of Fe_2_O_3_ nanoparticles individually [291].

##### Super-Paramagnetic Iron Oxide (SPION)

SPION is a modern approach to using magnetic particles; it induces local hyperthermia in the presence of a magnetic field [292] or may otherwise be protected in other nano-materials. Biofilms can be penetrated and degraded by Ag and Au and their magnetic effect [293,294,295].

#### 4.3.3. Organic-Based Nanomaterials

Organic-Based Nanomaterials involve nanomaterial primarily composed of organic material, with the exception of inorganic or carbon nanomaterial. The use of noncovalent (weak) molecular interactions tends to convert organic nanomaterial into desired structures, including liposomes, micelles, dendrimers, and polymeric NPs. Compared with inorganic materials, organic antibacterial materials are known to be less stable in nature, particularly at higher temperatures (Table 4).

##### Poly-ε-Lysine

A cationic homopeptide of L-lysine known as Poly-ε-lysine is effective against Gram-positive and Gram-negative bacteria. Additionally, it acts against spores of *B. subtilis, B. stearothermophilus,* and *B. coagulans* [296,297]. Some scientists have developed a technique to fight deadly antibiotic-resistant bacteria using nanocargos of gold nanoparticles (AuNPs) combined with ε-polylysine. These nanocargos were 15–20-fold more antibacterial compared with free poly-e-lysine when measured against carbapenem-resistant *Acinetobacter baumannii* (CRAB) referral strains and *Staphylococcus aureus* (MRSA) [298].

##### Quaternary Ammonium Compounds

The proven disinfectants are the quaternary ammonium compounds, including cetrimonium chloride, benzalkonium, and stearalkonium chloride. The antimicrobial action of quaternary ammonium compound relies on bash of the N-alkyl chain length and lipophilicity [299].

The mechanism of electrostatic interaction between a positively charged compound moieties and a negative charge bacterial membrane, followed by the integration of the hydrophobic tail compound into a hydrophobic membrane core where structural enzymes and proteins are denatured, culminated in initial encounters with the bacterial wall [300].

##### N–Halamine Compounds

N–Halamine complexes comprise one or multiple covalent bonds between nitrogen and halogen, which are common imide, amide, or amine group halogenations that provide stabilization and gradual emissions into the atmosphere of freely activated halogen species. These oxidizing halogenic agents facilitate the direct transfer of the active substance to the required biological locations or isolation from the aqueous medium to the halogen-free medium. These free, reactive halogens cause a microbial cell to inhibit or inactivate [301].

##### Polysiloxanes

Polysiloxanes, linear silicone oxide polymers, are yet another major type of polymer. Sauvet et al. synthesized a block of copolymers and statistical siloxanes having ammonium quaternary salt as a lateral substitution. These polymers show high antimicrobial action against *Escherichia coli* and *Staphylococcus aureus*. In block polymers or statistical copolymers, there was no significant difference in the activity [302].

##### Benzoic Acid, Phenol, and p-Hydroxy Benzoate Esters

Benzoic acid, phenol, and p-hydroxybenzoate esters are the most widely used disinfectants and preservatives. As monomers, the antibiotic activity of these compounds has already been identified. Attempts were made to combine them into a polymer backbone to synthesize the new, improved activity of antimicrobial polymers. Phydroxyphenyl acrylate has been more beneficial for both bacteria and fungi in a descriptive analysis of the antimicrobial activity of p-hydroxyphenyl acrylate, p-2-propane oxyphenol [303].

##### Quaternary Phosphonium or Sulfonium Groups

A broad spectrum of antimicrobial activity was found in quaternary ammonium compounds against both the Gram-positive and Gram-negative bacteria. Quaternary ammonium polyethylenimines (QPEI) provide a wide range of bacterial targets. When embedded in the different polymeric matrices, the polyamines have been shown to be an effective antimicrobial nanoparticles [304]. Polymers that have phosphonium quaternary or sulfonium groups have structures that are close to those of the compound ammonium quaternary group. Phosphonium-based polycationic biocides are more effective than quaternary ammonium salt polymers with respect to antimicrobial action. The NIPAAm and methacryloyloxyethyl trialkylphosphine chlorides experiments on water-soluble copolymer thermal sensitivity demonstrate that antimicrobial activity rises incrementally in polymer alkyl chain length and units of phosphonium in the polymer [305].

##### Triclosan

Triclosan is among the most common antimicrobial agents. A Triclosan solution was combined with water-based styrene–acrylate emulsion, and *Enterococcus faecalis* was tested. Depending on an agar diffusion test, triclosan liberation relies mainly on the solvent, either nonexistent or very sluggish in water and relatively fast with n-heptanes [306]. In addition to organic/aqua solutions, triclosan has been integrated, and PVA nanoparticles, which are water-dispersible, display higher antibacterial activity in relation to *Corynebacterium* [307].

##### Chitosan

The broad spectrum of antibacterial action has been identified in deacetylated chitin known as chitosan [308]. Chitosan nano-scale and its compounds have only recently been shown to possess antimicrobial action against microbes, viruses, and molds [309]. It is much more efficacious than bacteria against viral and fungal infections, and it has often been found to become more selective for Gram-positive than Gram-negative bacteria [306]. The molecular weight of Chitosan has a significant function in its antibacterial action, which also relies on variation in targeted bacteria’s cell wall: low molecular weight chitosan exhibits strong antimicrobial effects towards Gram (−) bacteria and high molecular weight towards Gram (+) [310]. Mechanisms through which chitosan works as an antimicrobial agent are described in multiple hypotheses:a.Adding it to the negatively loaded cell surface, inducing agglutination and even microbial cell permeation, allowing leaks of intracellular substances [309].b.Chitosan chelation characteristics used for the chelation of trace metals blocks the action of certain enzymes, causing cell death.c.Fungal chitosan produced through host hydrolytic enzymes from the fungal wall prevents RNA and protein synthesis [310].

Chitosan antimicrobial activities and water-soluble chitosan derivatives are observed to have significantly greater efficacy in combating bacterial membranes over others. Chitosan is a suitable, low-cost, and effective disinfectant for developing nations with a wide range of action and far less toxicity to mammalian cells [311].

#### 4.3.4. Composite-Based Nanomaterial

Composite nanomaterial consists of multiphase nanoparticles with a single nano-dimensional layer that can either mix nanoparticles with other nanoparticles or combine nanoparticles with smaller or larger materials (e.g., hybrid nanofibers) or more complex frameworks, such as metalorganic framework systems (Table 4).

##### Ceramic Matrix Nano-Composites (CMNC)

Nano-composites in the ceramic matrix are primarily Al_2_O_3_ or silicon carbide (SiC). Most of the research so far has shown that, after adding low-volume (approximately 10%) SiC particles of appropriate size and the heat pressure of the final mixture, the Al_2_O_3_ matrix has shown significant strengthening.

In a study, the coating system with a nano-TiO_2_ antimicrobial agent was used to prepare the antibacterial corrugating medium, and the antimicrobial efficacy was tested using the zone of inhibition approach. Furthermore, various concentrations of TiO_2_ antimicrobial agents have been observed in the mechanical properties of corrugating media, such as thickness, rigidity, bursting strength, tensile strength, and folding tolerance [312].

##### Metal Matrix Nanocomposites (MMNC)

Nanocomposites in a metal matrix (MMNC) refer to materials that comprise a matrix of a ductile metal or alloy that are implanted in a nano-sized reinforcing material. Metallic and ceramic materials combine these composites.

Metal nanocomposite active antimicrobial packages are developed by the incorporation of polymeric films of metal NPs. In nanocomposite antimicrobial mechanisms, the performance of NEs is effective primarily because of the high surface to volume proportion and large surface reactivity of the antimicrobial/metal oxide nano-sized particles, which enable them to more easily inactivate microbes [313]. Silver (Ag), gold (Au), zinc oxide (ZnO), silicon (SiO_2_), titanium dioxide (TiO_2_), alumina (Al_2_O_3_), and iron oxides (Fe_3_O_4_, Fe_2_O_3_) are some of metal and metal oxides nanomaterial widely used as antimicrobial agent.

##### Polymer Matrix Nano-Composites (PMNC)

The nanocomposites of the polymeric matrix are used extensively in heavy industry to make manufacturing simpler, lightweight, and flexible. They do, however, exhibit several drawbacks compared with metals and ceramics, such as low modulus and power.

Bio-nano-composite films built on poly (lactic acid) (PLA) and strengthened with nanoclay C30B (5.0 percent *w/w*) are infused with thymol and cinnamaldehyde active compounds at a high concentration of 11 to 17 percent *w*/*w*. The addition of active substances and nanoclay produces structural, thermal, mechanical, and antimicrobial properties to alter against specified Gram-negative and Gram-positive bacteria [314].
antibiotics-10-01473-t004_Table 4Table 4Types of Nanomaterials with their efficacy against bacteria.NanoparticlesParticle Size(nm)Targeted Bacteria and Antibiotic ResistanceAntibacterial MechanismsFactors Affecting Antimicrobial ActivityReferencesInorganic NanomaterialsFe_2_O_3_ NP1–100MRSA*K. pneumoniae*MDR *E. coli*Disruption of cell walls through ROSDispersibilityHigh chemical activityAir oxidation leading to magnetismAggregation occurs[315][316]Ag NP1–100MDR *Escherichia coli**Staphylococcus epidermidis*MRSA*Pseudomonasaeruginosa**Vancomycin-resistant Enterococcus**A. baumannii, carbapenemresistant**P. aeruginosa Carbapenemandpolymyxin B-resistant**Carbapenem-resistant**Enterobacteriaceae**Klebsiellapneumoniae**Extended-spectrum betalactamase-producing organisms*Lipid peroxidationIntercalationbetween DNA basesROS generationInhibition of cell wall synthesisInhibition of cytochromes in the electron transport chainRibosome destabilizationDissipationof proton gradient resulting in lysisIncrease in membrane permeabilityCell surface binding which causes lipid and protein deteriorationBacterial membrane disintegrationShapeParticle size[317][315][318][316]ZnO NP10–100*K. pneumoniae,**Enterobacteraerogenes,**ESBL-producing E. coli*MRSA*K. pneumonia**E. coli**Klebsiellaoxytoca*Lipid and protein damageAdsorption to cell surfaceROS production, disruption of membraneConcentrationParticle size[319][315][318]Cu NP2–350*A. baumannii*MDR *E. coli*DNA degradationROS generation,Cell membrane potential dissipationProtein oxidationPeroxidation of lipidConcentrationParticle size[320][317][318][316].Au NP1–100MRSAbacterial membrane disruptionRespiratory chain damage,Reduced activity of ATPaseThe generation of cell wall apertures.Loss of membrane potentialDecline in tRNA binding to ribosome subunitParticle sizeRoughness[321][317][316]TiO_2_ NP30–45*S. aureus**E. coli**Enterococcus faecium**P. aeruginosa*Adsorption to thecell surfaceROS generationParticle sizeShapeCrystal structure[315][318]Si NP20–400MRSADisruption of cell walls through ROSParticle sizeStabilityShape[317][316]MgO NP15–100*S. aureus**E. coli*AlkalineeffectROS generationElectrostatic interactionLipid peroxidationpHParticle sizeConcentration[315]Al NP10–100*E. coli*Disruption of cell walls through ROSParticle size[315][318]SPIONS15–25*S. aureus**E. coli*NO releaseProduction of ROS.Particle size[322]Organic NanomaterialsPoly-ε- lysine 1–100*S. cerevisiae**B. subtilis**B. stearothermophilus**B. coagulans**E. coli*Disrupt the cell wall and membrane integrity.Destroy cell membranes or cell wallsParticle sizeConcentration[323]Chitosan200*S. aureus**E. coli*Loss of permeability of membranepHConcentration[324]Quaternary ammonium compounds 1–100*Pseudomonas**Pseudoalteromonas**Erwinia**Enterobacter*Interfere with the function of the cell membraneLysis, or destruction of the cellAffects DNAROS releaseParticle sizeConcentration[325]N-halamine compounds1–10*S. aureus**P. aeruginosa*Interfere with the function of the cell membraneComplete inactivation of the bacteriaConcentration[326]Quaternary bis-phosphonium and ammonium1–100*S. aureus**S. epidermidis**B. subtilis**E. coli*Inhibits the growth of bacteriadisruption of the cell division mechanismsCrystal structureParticle size[327]Carbon-Based NanomaterialsFullerenes200*E. coli**Bacillus subtilis*Outer membrane damageROS generationParticle sizeShape[328]CNTs1–100*E. coli**Streptococcus spp.**S. eidensis**E. faecium**A. baumannii**Y. pestis**B. epacia**K. pneumonia**S. enteric*Respiratory chain damageEnergy metabolism inhibitionPhysical interactionsSevere damage to the bacterial membraneParticle size[329][330]GrapheneOxide NPs12*E. coli**K. pneumoniae**E. faecalis**P. aeruginosa,**S. aureus*Methicillin-resistantMDRMerge antibiotics with the NIR treatment.UV irradiation contributes to ROS production.Several toxic pathways.[331][332]Composite-Based NanomaterialsCeramic Matrix Nano-composites1–100*S. aureus**E. coli*High antimicrobial effectInhibit the bacterial growthPhysical interactionParticle sizeHeat pressure[333]Metal Matrix Nano-composites1–100*A. baumannii**S. aureus**E. coli**K. pneumonia*Inhibit the bacterial growthPhysical interactionformation of irregular pores in the outer membrane of bacteriaParticle sizeDepends on the content in the medium[334]Polymer Matrix Nano-composites1–100*A. baumannii**S. aureus**E. coli**K. pneumonia**MDR*Inhibit the bacterial growthPhysical interactionDepends on the content in the medium[335]


### 4.4. Mechanism of Action of Nanoparticles

The use of nanoparticles to fight bacteria is a fairly modern approach that may provide a viable solution to the crisis of global antibiotic resistance. Theoretically, it would be difficult for microbes to establish a resistance to several destroying pathways utilizing nanoparticles because of their multiple-target and action capability. 

Many experiments have been conducted to test the metal-based nanoparticles’ mechanism of action. The nanoparticles are known for their strong antibacterial activity against a wide array of pharmaco-resistant species in silver and silver oxide. They use several mechanisms to work against bacteria, rendering them most efficient. The silver nanoparticles work by generating vast amounts of silver ions, which influence the permeability of the cell membrane and help to inhibit energy transfer through the transport chain of electrons. In addition, microbial cell damage to DNA [336,337,338] has also been identified. Zinc oxide nanoparticles are another type of nanoparticle which grows in the cells and emit Zn^+2^ ions there. These involve hydrogen peroxide production and destruction of the cell membranes [339,340]. Titanium dioxide nanoparticles participate in reactive oxygen production and subsequently influence the integrity of the cell membrane [341,342,343]. In contrast to metal nanoparticles, numerous nanomaterials dependent on polymers, liposomes, or carbons, each of which has its own mechanisms of action, are often used for fighting pathogens. Nanoparticles with chitosan work by boost permeability and breakout of the membrane. They as well engage within enzyme inactivation of the microbial machinery [227,309]. Carbon nanotubes act primarily in producing and oxidative degradation of the cell membrane, lipid, and proteins through reactive oxygen species [344]. A relatively recent family of nanoparticles that involves fullerenes works by rising neutrophil infiltration and is active in cell membrane disruption [345,346]. These various action pathways are ways in which nanoparticles can successfully attack the microbial machinery (Figure 5).

### 4.5. Drug Release Kinetics of Nanoantibiotics

The effectiveness and efficiency of a nano-formulation filled with medicines depend mainly upon the kinetic release. For maximum effectiveness, it is essential to have medicines released for nano-carrier slow before reaching the site of action to prevent medication loss [347,348,349]. This procedure must be tested and continuously monitored to guarantee optimal and safe activity at the site and to avoid off-site behavior and possible subsequent adverse reactions. The drug release kinetics is the key factor in the effect nanoparticles, designed as antibiotic nano-carriers, will have on medication delivery [350,351]. Popular techniques for evaluating the release of NPs include dynamic ultracentrifugation, ultrafiltration, dialysis, and others. Dynamic dialysis is often preferred since additional phases in the isolation and release of medication from NPs are omitted at different kinetic periods. Furthermore, the exterior stress applied in several other approaches during the separation often reduces the efficiency of those methods [352]. Commonly, a broad range of technics is used to include antibiotics, such as liposomes, nano-emulsion, polymeric NPs, micelle systems, dendrimers, and strong lipid NPs, in targeted, regulated, and appropriate manner [353]. The drug release kinetics for polymeric NPs depends largely on the following:i.Surface-bound or adsorbed product desorptionii.Drugs diffusion from polymeric NPsiii.Polymeric NP erosion and a cumulative erosion/diffusion effect

Unless the rate of membrane degradation becomes more significant than the product diffusion from the polymer layer, diffusion happens as a consequence of drug exposure. The initial “burst-free release” effect is, typically, a drug release from polymeric NPs, which implies the surface release of a compound bound or adsorbed in contrast with that used within the matrix [354]. Compared with polymeric NPs, the release of the drug from liposome depends largely on the following:i.Lipid membrane compositionii.Nature of drug involvediii.The percentage of drug’s permeabilityiv.Environmental issues to consider such as temperature, pH, or exterior stimulations like degradation of the enzymes, ultrasound, or interaction of the proteins [91,92,93,94]

A significant focus has been placed on designing nanoparticles with improved drug delivery mechanisms. Several techniques for regulation of product release kinetics have been implemented, including the use of anionic gemini surfactant and covalent bonds, which can disrupt the pH of polymer NPs within an acidic environment [355]. In order to produce new nanoparticles with improved product supply and antibacterial action, the filling of the drug with polyelectrolytes is now an emergent trend [356]. Numerous experiments have examined nanoparticle drug loading and release kinetics for directed and controlled release. The chitosan-magnetic iron oxide NPs (Strep-CS-MNP) packed with streptomycin exhibited a significant burst discharge preceded by controlled drug release kinetics based on the physical mixture of chitosan–streptomycin, as well as magnetic iron oxide NPs, implicit the need for monitored drug release pathways. The analysis found that particles observed kinetics with a correlation (R) valuation of 0.9863 in the quasi-2nd order. The analysis found that Strep-CS-MNP had greater antibacterial activity than Streptomycin only toward MRSA. Other studies have found an improved production of the pH-responsive vancomycin-loaded chitosan NP at pH 6.5 relative to pH 7.4. The treatment further revealed that the risk of MRSA in a design of skin infection of the mouse was 8-fold reduced compared with those handled with pure vancomycin [353].

Newly published research indicates that the 90% loaded release of the drug around 2.5 h after first-order release kinetics has been generated by single-walled carbon nanotubes (SWCNTs). The drug demonstrated 16 times greater efficacy in contrast to ciprofloxacin against *P. Aeruginosa and S. aureus* and eight times against *E. coli* [357]. Different structures were used together to incorporate metal and antibiotic, antibacterial results. In yet another study, 120 h of vancomycin release profile with high antibacterial impact toward MRSA were shown in vancomycin-loaded aragonite NP. Such NPs, with a broad release profile, will serve as a strong local antibiotic supply mechanism for osteomyelitis treatment [355]. Research further showed the design of well-ordered, biocompatible Cu-doped MSNs with an extensive specific surface area, subsequently encapsulated with tetracycline (TET) species and coated with ultrasmall silver nanoparticles-stabilized polyethyleneimine (PEI-SNP) complex layer for combating antibiotic bacterial resistance. Outcomes from the bioassay showed that SNPs increased the antibacterial potential of the strain of MDR *E. coli* substantially by cell membrane sensitization and enhanced intracellular access to nanoscale containers for pH-caused drug cargo transport. In addition, the degradation of MDR bacteria was aided by massive ROS levels owing to Cu species [358]. An effective approach has been proposed for the delivery of antimicrobial agents, in particular at acidic pH. Nanoparticles were immobilized with silver–indole-3 acetic acid hydrazide (IAAH–Ag) complexes via a pH-sensitive hydrazone bond, which acts as a model drug. The synthesized IAAH-Ag pH-sensitive complex was shown to have a robust antimicrobial ability in the planktonic and biofilm states towards multi-drug-resistant bacterial isolates. Nanoconjugates have been shown to be of strong effectiveness in the treatment of bacterial mice infection [359].

### 4.6. Synthesis of Nanoparticles

The special features of biologically synthesized NPs are more significant than nanomaterial developed by chemical and physical methods. Nanoparticles may be synthesized physically/chemically [360]. These processes may pose a variety of issues, such as the use of dangerous chemicals, the generation of hazardous by-products, and geometrical errors [361]. Chemical processes typically consist of using one or more organic compounds or molecules, which aid particle toxicity and sensitivity and could create a risk to the environment and human health [361].

Green synthesis particulates are clearly distinguished from physical–chemical ones. The bottom-up method of green synthesis is similar to a chemical reduction in which expensive chemical-reducing compound is substituted with extracts of natural substances, such as fruits or tree leaves for metal or metal oxides NP synthesis, or by using different biological organisms. The nanoparticles produced through green synthesis are known as ‘biogenic nanoparticles’. Biological species have enormous production potential for NPs. Biogenic reductions of metal precursors to link NPs are environment friendly [362], safe [363], chemical-free [364,365], and can be used in large amounts. In addition, expensive metals like gold or silver can be recycled through the production of biological nanoparticles. These metals have limited sources, and their costs fluctuate [366]. NPs’ can easily communicate with other biological molecules, a capacity that improves antimicrobial behavior by enhancing the interaction mechanism of microorganisms, primarily with sugars enzymes, proteins, and even the entire cells [367]. The biological composition of the biogenic NPs allows for smooth drug isolation or up-concentration through centrifugation of biogenic NPs from the reaction media [368].

For biogenic NP synthesis, biological removal of metal precursors generally occurs in vitro or in vivo. Carbohydrates, enzymes, and plant chemicals such as flavonoids, phenolics, terpenoids, cofactors, and others, are thus mostly reduced and stabilized [369,370]. It has been suggested that fungi, bacteria, algae, plants, and yeast produce biogenic NPs in-vivo [371,372,373]. Biological extracts mostly used in in vitro synthesis include the purification of organic agents and mixing them in a controlled manner for the related metal precursor into an aqueous solution. This spontaneous reaction takes place at around room temperature [374], but it is often vital to involve stirring and heating [375]. These items are environmentally safe for the processing of biogenic NPs and their waste materials [376].

#### 4.6.1. Green Synthesis of Nanoparticles

The green synthesis of NPs will typically adopt either the bottom-up or top-down method (Figure 6). NPs are generated by reducing the size and via different chemical and physical techniques in the top-down approach [377]. NPs are formed by smaller structures, such as molecules and atoms, with the key effect on oxidation/reduction in the bottom-up synthesis. NPs with homogenous chemical characteristics and limited errors are extracted in this sustainable and green pathway. Plant extracts and microorganisms are commonly used in the biological method of NPs production (Figure 7) [378,379,380]. The choice of the suitable species or extracts will consider the specific features, such as phytochemical substances, biochemical processes, enzyme functions, circumstances of cell development, and optimum reaction [381].

##### Fungi

Fungi are secondary metabolite contributors and active biomolecules, which are very important for NP synthesis. Fungal species such as *Foxysporum* produce proteins, polymers, and enzymes that willingly support the development of metal NPs [382]. These components enhance NP outcome and stability. In studies, some fungal organisms were reported to generate NPs with traces of extracellular amino acid. For example, the surface of the yeast includes glutamic acid and aspartic acid, which, in the presence of sufficient light, transforms silver into silver ions [383]. Ahmad et al. (2003) have found that fungal species such as *Foxysporum* have cytosol reductase enzymes that, in the presence of NADH^+^ reduction element, reduce silver from silver ions [381]. The phytochelatin class is strongly able to suppress silver ions in silver metal [384], which are present primarily in fungus. The population of fungal *Coriolusversicolor* was used by Sanghi and Verma (2009) for the silver NP synthesis (Ag NPs). In this study, FTIR data revealed the existence of the hydroxyl group in fungal mycelium, which contributes electrons to the silver ion and decreases them to transform Ag NPs into bare metal. In addition, it is reported that aromatic and aliphatic amines and other proteins in the fungal extract used to maintain Ag-NPs developed as a capping agent. It was further shown how silver powder became stabilized by the amide coupling of protein [385]. The role in the stabilization and capping of Ag NPs by SH group comprising fungal extract protein has also been documented by Tan et al. (2002) [386]. Das et al. (2009) utilized *Roryzae* mycelia for synthesizing gold (Au) nanoparticles by reducing the in situ of acidic chloroauric acid (HAuCl_4_) (pH 3) [387]. *Verticillium* fungus is indeed an excellent mediator for silver NPs synthesis. Biomass of fungus has generally been discovered to produce intracellular NPs on AgNO_3_ treatment in an acidic medium (pH 5.5–6) [388] (Table 5).

##### Bacteria

NP synthesis aided by bacteria is produced by two methods: intracellular and extracellular. Less time-consuming, extracellular synthesis of NPs is more advantageous than the intracellular system, provided that no downstream process is needed in order to gather NPs from organisms [396,397]. On the inside of the cell, the bacteria contain reductase enzyme, which catalyzes metal ion decrease into metal NPs. *Dradioduran* species have high antioxidant activity and are highly resistant to oxidative stress and radiation. This makes it suitable to use Au NPs in green synthesis from their ionic form (Table 6).

*Leptothrix* bacteria were used to synthesize AU-NPs by reducing gold salt on an aqueous medium in a new analysis by Kunoh Tatsuki et al. (2017). It has been documented that guanine RNA molecules and 2-deoxy guanosine residues minimize gold salt (Figure 6).

##### Plant

Plant-aided NP synthesis is far more effective than microbial synthesis. Plants possess numerous biochemicals (e.g., polyphenols) and many metabolites, which can act as stabilizers and reduce the synthesis factor of biogenic NPs. The synthesis of NPs in plants is environmentally sustainable and cost-effective. Plant-based NPs have been shown to be somewhat more stable than microbial and fungal NPs [396]. The controlled plant synthesis of the NPs is divided into three groups; phytochemical, intracellular, and extracellular materials. When plant extract is used as the raw material for the synthesis of NPs, the extracellular method is used. Intracellular synthesis of NPs occurs within plant tissue cells utilizing cellular enzymes. After synthesis, by destroying the plant cell wall, the NPs are restored. Synthesis of plant extract NP is relatively inexpensive and results in higher yields because more plant extract phytochemicals can stabilize or transform ions of metal into metal NPs [404]. Phytochemical controlled NP synthesis is not a commonly used method, as it involves understanding the specific phytochemical required for balanced NP synthesis [405]. Shakeel Ahmed et al. (2015) produced Ag NPs in a spherical form via *A. indica* leaves extract. In the FTIR study, the flavonoids and phytochemical compounds of the plant extract were found to act as reducers and stabilizers during the synthesis of NP. There was a high antibiotic response in these NPs [406]. Suman et al. (2013) isolated Au NPs through *M. citrifolia* root extract that had also been reported to be antimicrobial in nature [407] (Table 7).

#### 4.6.2. Purification of Nanoparticles Extracted Biologically

Purification of nanoparticles after the synthesis is quite important before using them in any sort of application (Figure 6). Scientists have increasingly used the centrifugation process to purify nanoparticles since the procedure is simple and time-effective. Therefore, for the isolation and purification of metallic nanoparticles produced biologically, in order to isolate unreacted bioactive molecules, frequent washing and high-speed centrifugation are performed [405]. However, the process has several drawbacks, such as centrifugation which may induce NP agglomeration. Because of the disunion of the passivating agent from NPs and a transition in the NP’s underlying properties, NP destabilization occurs. The process of dialysis, using the exact cutting of the membrane, is another common form of purification. The dialysis membrane can quickly transfer tiny organic molecules contained in plant extracts, although organic molecules found as surface-passivizing agents stay in place and are connected with NPs in the dialysis membrane. This filtration process requires time which normally takes more than 24 h. However, for bio-fabricated nanoparticles, the diafiltration method is not used as it is insoluble in soil. The application of external magnetic power effectively splits up the magnetic nanoparticles such as Fe_2_O_3_ and Fe_3_O_4_. Moreover, it is always challenging to extract closely bound biomolecules from the nanoparticle layer (Figure 6).

#### 4.6.3. Nanoparticles Coating for Antibacterial Activity

Some researchers have obtained four types of antibacterial materials and coatings during the last decade. The first types of antibacterial coatings are those to which bacteria do not show a natural tendency to attach. Frequently, this form of coating is focused on hydrophilic polymers like polyethylene glycol (PEG), oxazolines, radicals in nitroxide, and chlorinated plasma polymers [419,420,421]. The second type of coatings or compounds that are modified can destroy bacteria while in contact. Studies have shown that an RNH^4+^ bonded nitrogen concentration of 4.18% and a surface potential of +120.4 mV is needed for the killing of Escherichia coli efficiently [422,423], with a carefully planned amount density gradient of QAC. Figure 8 presents strategies that provide excellent approaches to avoid the colonization of bacteria on the device surface while the bacteria that have penetrated the implantation site are not removed. An exposed wound is therefore infected by opportunistic pathogens. Coatings or materials that release antibacterial agents have been produced for pathogen neutralization. A significant range of antibacterial substances, including classical antibiotics [424,425], nitrous oxide [426,427], antibacterial polymers, and peptides, may be issued under this strategy [428,429,430]. The last category consists of coatings and materials that trigger antibacterial material only when the product has been infected or engulfed by photogenic bacteria (Figure 8) [431,432,433].

### 4.7. Factors Influencing the Synthesis of Various NPs

Three main factors influence the synthesis of various NPs. They are temperature, pH, and reaction time (Figure 9).

#### 4.7.1. Temperature

Significant research is being currently being carried out worldwide on the question of temperature regulation in NPs. Temperature is one of the most critical parameters that affect the morphology and synthesis of the NPs. The different forms, dimensions, and synthetic structure of NPs are temperature-dependent (triangles, platelets of octahedral, spherical, and rod-like). The rate of the reaction, as well as the formation of nuclear centers, tends to increase with the increase in temperature [434,435,436]. Sneha et al. (2010) analyzed the temperature effects on the morphology of Piper leaf extract synthesized Au NPs [437]. They found triangular NP types at 20 °C, while octahedral NPs of 5–500 nm were produced at temperatures from 30 to 40 °C. Therefore, the size distribution of the NPs was much more appropriate and spherical in the form at increased temperature (50–60 °C). Iravani and Zolfaghari (2013) [438] documented the biological development of Ag NPs via *P. eldarica* bark extracts under different temperatures. The synthesis was conducted at 25, 50, 100, and 150 °C. Triangular NP types were found at 20 °C, while octahedral NPs of 5–500 nm were produced at temperatures from 30 to 40 °C. The NPs were produced at 20 °C. Electron micrographs scan of NPs confirmed this result. In the synthesis of the AgNPs using biocompatible polymer PEG [439], the effect and temperature of atmospheric oxidation have been investigated by Fleitas-Salazar et al. (2017). The analysis showed that PEG had a tendency towards decreasing silver salt at 100 °C. It was also suggested that the functional PEG groups strongly interacting with Ag molecules at 100 °C resulted in a more balanced and stabilized Ag-NP structure. At 60 °C, Ag ions were reduced by hydroxyl group oxidations found in PEG. The researchers have thus confirmed that there are many methods for Ag NP synthesis operating at different temperatures. The synthesis procedure for AU NPs in aqueous poly (ethylene oxide)-poly (propylene oxide) solution at various temperature levels was developed by Islam et al. (2011). Their findings revealed that the NPs are regulated by changes in polymer morphology occurring at relatively low temperatures, distribution, and size. However, modification in polymer chemistry is regulated at higher temperatures [440]. The impact of temperatures upon the encapsulation and development of Au NPs was studied by Tans et al. (2015) using the standard PNIPAm/PEI. The TEM analysis showed that the optimum encapsulation of Au NPs, and that of stable Au-PNIPAm/PEI composite particle production, is between 25–30 °C, with homogeneous particle distribution across the template. The encapsulation at a lower temperature (15 °C) was relatively poor [441]. In another review of the synthesis of cobalt ferrite NPs doped by Manganese [442], the number of NPs was shown to rise with the increase in temperature.

#### 4.7.2. pH

The pH of the reaction plays a crucial role in forming NPs. The formation of nuclear centers is also regulated by the pH-like temperature. With the increase in pH, there is a parallel rise in the number of nuclear sites, hence the growing number of metal NPs. Several studies on the effectiveness of pH in developing the morphology and scale of NPs have been conducted. Armendariz et al. (2004) have observed the synthesis of *A. sativa* Au NPs at various pH levels. The lower pH (pH 2) revealed fewer NPs but a much greater NPs size (25–85 nm). They indicated that Au NPs do not generate any new nuclear centers at a lower pH but instead had a tendency to aggregate into larger NPs [443]. In contrast, small NPs were developed with a slightly higher pH (pH 3–4). Fan et al. performed an additional analysis of the synthesis of pH-dependent NPs. They monitored the release of NIPAm poly-(N-isopropylacrylamide)/chitosan NPs filled with Camptothecin into the tumor and noted that when the NIPAm and chitosan ratio is 4:1 (*w/w*), the loaded NPs are released to the target. At pH 6.8 the rate of release of Camptothecin was considered to be ideal. However, they noted that the release rate decreased as the pH decreased or increased to the 37 °C temperature. Okitsu et al. (2009) [444] performed further experimental analysis to examine the impact of pH on dimensions of the average size of the gold nanorods and concluded that the ratio and the size decreased with an increase in pH.

#### 4.7.3. Reaction Time

The reaction time, along with the temperature and pH, is considered an essential factor affecting the NP morphology. Karade et al. (2018) [445] conducted an analysis of the impact of the reaction time on magnetic NPs. Fe NPs have been synthesized with green tea extract using Ferric nitrate solution. It was reported that the reaction time affected both the structural and magnetic properties of magnetic NPs. The size of the particles enlarges from 7.5–12 nm as the reaction time rises. Increasing reaction time has also been observed to increase the magnetic saturation (Ms) of NPs. Additionally, the impact of the duration of reaction on cadmium selenide NP particle size has been studied [446]. The estimated particle sizes at 4, 8, 12, and 16 h reaction times were determined to be 15.8, 10.5, 6.7, and 111.7 nm, respectively. The size of cadmium selenide NPs decreased with an increase in reaction time. It was reported that the unusual size increase at 16 h resulted from particle accumulation. Furthermore, Flor et al. (2004) studied the impact of response time on the particle sizes of ZnO and Cerium doped ZnO [447]. The study indicated the linear increase in particle size with reaction time rise. It was found that at constant periods of reaction, cerium doped particle sizes are greater than standard ZnO. Finally, the impact of the reaction time on stability, size, and reduction of AuNPs synthesized from oil palm oil extracted (*E. guineensis*) was examined, and it was observed that with the rise in reaction time, the reduction rate of AgNPs was also increased.

### 4.8. Factors Influencing the Activity of Nanoparticles

Nanoparticle behavior toward the microbes (bacteria and fungi) may be caused and influenced by various factors. Here, we identified several that could, theoretically, affect nanoparticle antimicrobial behavior. Chemical structure or size, composition, the concentration of nanoparticles, along with their form, target microorganism nanoparticles are acting against, and the effect of photoactivation are addressed below (Figure 10).

#### 4.8.1. Chemical Composition of Nanoparticle

Chemical Composition is the crucial feature of the antibacterial mechanism in nanostructures. Many studies have focused on the choice of a substance suitable for nanoparticle synthesis and the potential effects such choices would entail for the antibacterial mechanisms [281,448]. Analysis revealed the significance of the chemical compositions of nanoparticles and the way nanoparticles are sprayed against bacterial cells. There were substantial effects in copper nanoparticles compared with iron, size 30–40 nm and 30–70 nm, respectively, of ten drug-resistant *S. aureus*. Recent research interest has turned to the combined production of nanoparticles with the objective of achieving chemical and antibacterial activities.

#### 4.8.2. Shape of Nanoparticles

Nanoparticle shape has been shown to affect the effectiveness of several antibacterial nanoparticles. The shape-dependent behavior of silver nanoparticles was defined by three distinct shapes (sphere, elongated, and truncated triangular silver nano-plates). Analysis of *E. coli* activity was based on the percentage of the active facets. Increased antimicrobial activity with more facets [111] with high atomic density facets was observed [224].

#### 4.8.3. The Target Organisms

The micro-organisms targeted by the nanoparticle have significant effects on its activities. A higher percentage of AgNP activity towards Gram-negative rods than Gram-positive cocci and higher tolerance to *E. coli* than to *S. aureus* for silver nanoparticles are good examples. The results were linked to peptidoglycan that is absent in mammalian cells in the *S. aureus* cell wall. The cell walls of both Gram-negative and Gram-positive bacteria, in contrast, were disordered by ZnO NPs [449,450]. In ZnO NPs, the operation against *S. aureus* was higher than that against *E. coli* [263,451]. These findings demonstrate the target-dependent nature of the relationship between antibacterial activity and nanoparticles.

#### 4.8.4. The Photoactivation

Photoactivation is considered the most relevant parameter of the antibacterial behavior of the nanoparticle. When exposed to UV radiation, the TiO_2_ NPs were considerably more active against *E. coli* [452]. TiO_2_ nanoparticles with exposure to standard laboratory lightning exhibited a 20% increase in growth control properties, and ZnO nanoparticles, when exposed to visible light and UV radiation resulted, also showed increased activity [453].

## 5. Characterization of Nanoparticles

After the synthesis of the NPs, a range of methodologies are used to evaluate their conformational specifications for size, form, homogeneity dispersity, and morphology. Dynamic Light Scattering (DLS), UV-Vis Absorption Spectroscopy, X-Ray Diffraction (XRD), Electron Microscopy Transmission (TEM), Energy Dispersive X-ray Analysis (EDAX), and Scanning Electron Microscopy (SEM) are methods most frequently used to characterize NPs (Figure 11).

The UV—Vis absorption spectra are used for the aqueous suspension of NPs in size and form [454]. To identify the NPs of approximately 2 to 100 nm [455], wavelengths from 300 to 800 nm are usually used. For instance, UV—Vis emission spectra of Aloe Vera extract synthesized ZnO particles display high UV spectrum absorption with a higher wavelength of 358 nm to 375 nm because of their membrane Plasmon resonance [456].

The SEM and TEM describe the morphology and scale of NPs [457]. ZnO-NP (25 to 55 nm) seen in the electron microscopy study is compatible with the XRD [458]. The green synthesized carbon nanotubes were analyzed by SEM and TEM, which were completely coated with polyaniline layers [459]. TiO_2_ particles were commonly spherically agglomerated within the range of 10 to 30 nm in the TEM examination. In addition, a crystalline structure was shown by Selected Area Electron Diffraction (SAED) [460].

XRD contains data regarding symmetry, sizes, and the state of metallic NP detection [461]. X-rays penetrate nanomaterials. The division sequence collected is correlated with structural knowledge requirements. XRD peaks (2 h) at 28.51, 33.06, and 47.42 angles of 111, 200, and 220 planes, respectively, and normal separation peaks at the front-and-center cubic step of the CeO_2_ NPs [462]. The XRD analysis confirmed the existence in the Scherer equation (Elango and Roopan 2015) of the crystalline patterns of PbNPs and the average particle size of 47 nm.

FTIR Spectroscopy is designed to identify different types of functional groups or metabolites that may contribute to the reduction and stabilization of NPs at the surface of NPs [378]. The functional group bands observed at 3450, 3266, and 2932 cm^−1^ have been allocated to stretching the amines frequencies, O–H to stretch alcohols, and C–H to extend the alkanes for NPs using Aloe-Vera extracts. ZnO is allocated for the peaks in the region of 600–400 cm^−1^ [463]. In 1648, 1535, 1450, and 1019 cm^−1^ and 1450 cm^−1^ peaks of carboxyl ions, the FTIR range of AgNP synthesized using the *Solanumtorvum* leaf extract was reportedly responsible for stabilization of the Ag NPs [464].

DLS and EDAX are used to study the flow of size of liquids and essential components of NPs accordingly [465,466,467].

## 6. Comparison of Antibiotics with Nanoparticles

Although antibiotics have been important historically, nanoparticles are increasingly being used in emerging medical research and application. If the efficacy and potential of antibiotics and nanoparticles are to be compared, their combined action will prove to be more significant than that of either one alone. Some comparisons are offered below in Table 8.

## 7. Antimicrobials and Nanoparticles in Combination

As already stated, NPs do not only defend themselves against bacterial and microbial resistance but can also be the “medium and carrier” for antibiotics. The methodologies of NP-based drugs vary based on the pathways discussed earlier.

The essential features of NPs as a carrier for the delivery of antibiotics, in contrast with conventional delivery systems, are:Size: the ultra-small and customizable size of NPs is ideal for performing antimicrobial activities and combating intracellular bacteria [467,468].Protection: NP carriers can help protect the drugs from resistance by target bacteria by increasing the serum levels of antibiotics [280,469]Precision and Safety: NP carriers can help find a contaminated region with antibiotics and eliminate systemic adverse reactions [470,471].Controllability: safe and controllable antibiotic release can be flexibly achieved [472,473,474].Combination: the same NP may be used to combine several antibiotics or drugs, and NPs can be combined with others to enhance the antimicrobial properties of the antibacterial agents [282,472,476]

Recently, researchers have combined the gold and silver nanoparticles with Ampicillin. They reported that silver nanoparticles have an inherent ability to combat the microorganism, although gold nanoparticles have an antimicrobial effect only when in surface communication with ampicillin. Broad-spectrum action against the Gram-negative bacteria and Gram-positive bacteria is accomplished by the ampicillin-functioning of silver and gold nanoparticles. Silver and gold combined with ampicillin are very effective in treating bacteria resistant to antibiotics [477].

Any effective antibiotic, including certain penicillins, can be recovered through the use of antibiotics with metallic nanoparticles [245,252,261,267]. In addition, the combined use of nanoparticles with antibiotics or other antimicrobials allows these agents to decrease the toxicity in human cells [269]. The majority of research is dedicated to the study of the interactions of nanoparticles and antibiotics and their various combinations, especially with b-lactams (ampicillin, amoxicillin) and glycopeptides (vancomycin), which have shown promising enhancing effects in vitro. These effects are much more likely due to the increased penetration of these nanoparticles in the cell wall. While the interaction of other metallic nanoparticles with antibiotics has still not been thoroughly investigated, ZnO and TiO_2_ nanoparticles have shown, with limited evidence that they can respond to efflux pumps that cause the resistance of many clinically important antibiotics, such as fluoroquinolones and many more [478].

## 8. Antimicrobial Applications of Nanoparticles on Animal Model

Although there are currently several applications of the nanoparticles, only a few studies have been conducted in the context of antimicrobial activity. Some of the antimicrobial applications of nanoparticles on different animal models are shown in Table 9.

## 9. Challenges for Nanoparticle

The benefits of nanotechnology in numerous fields, including medicine, are well-established, but the potential implications or harmful effects of these nano-sized particles are not well-explained. In the medical context, as an alternative antibiotic, a whole range of functions and relationships of the nanoparticles can be found [483]. A number of groundbreaking studies are underway in nanotechnology, while the critics of these new technologies find it incredibly challenging to reconcile the side of nanoparticles with their reported strengths. As nanotechnology development is gaining momentum, concerns have been raised about its safety, particularly in medicine [484]. Nano-formulation plays an essential role in the development of nano-based drugs and thus faces regulatory issues relating to medicines. To design a new nanodrug based on the form of drugs already existing, specific regulatory manufacturing criteria must be provided and rigorously adhered to throughout drug production. A shorter approval process is pursued for nanodrugs developed from previously licensed micro-formulations; however, when a novel product is formulated, the paths of assessment and approval are more stringent [485]. Manufacturers must follow the guidance of the FDA (Current Appropriate Manu—Invoicing Practices) and the Quality Control Regulations for the production of new Nano drugs.

Nanoparticles have significant benefits and advancements in the management of infectious diseases over conventional antibiotics; however, their delivery technique is still difficult in clinical applications. For proper therapeutic results and effective clinical application, an assessment of the potential nanoparticle interactions with tissues, organs, or cells, their doses, and potential mode of administration, is needed [486]. Nanoparticle toxicity is necessary for effective clinical translation to take place. [487]. However, intravenous (I.V.) NP administration can lead to NPs accumulating in bone marrow, spleen, liver, and lung [488]. Due to their small size and effective cell absorption, inhaled NPs can enter the liver, brain, spleen, heart, and lung [489]. Moreover, the toxicity of several nanoparticles is not well known [483]. NP therapeutic administration can produce nanotoxicity of multiorganisms. All toxic cases show NPs association with oxidative stress-inducing cells contributing to hepatotoxicity and pulmonary toxicity [490,491]. Suggested metabolic modifications such as mitochondrial dysfunction, decreased ketogenesis, beta-oxidation of fatty acids, and glycolysis contribute to hepatotoxicity and nephrotoxicity [492]. While preexisting in vitro methods have certain advantages, there are no popular practices for universal NP dosing [484,488], as nanoparticles demonstrate size-specific action. To solve these problems, additional characterization methods are required [490]. More recent work mainly focuses on delivering targeted bacteria [493,494].

## 10. Conclusions

Antimicrobial resistance has posed an unprecedented global threat to human life. Conventional antibiotics are losing their potency against the ever-evolving multiple drug-resistant pathogens. In search for a viable alternative strategy, a nanotechnology-offers promise and nanotechnology-based drug delivery device for the production of future nano-antibiotics (nAbts) is seen as a weapon in the 21st century technological revolution. To evolve novel pharmaceutical drugs over time, the newly developed field of nanotechnology is still in its infancy and requires significant effort and investment. Compared with traditional antibiotics, nanoparticles can have numerous benefits, such as longevity, absorption, controlled release, distribution, and delivery.

Furthermore, for the resistant, antimicrobial environment, nanoparticles can be cost-effective and ecological, as well as flexible. Nanoparticles have distinct and well-defined physical and chemical characteristics that can be customized for desirable purposes. Moreover, owing to the excessive volumetrically surface area, they have a strong antimicrobial performance that gives them an advantage over their chemical counterparts facing drug resistance challenges. The advances in nanotechnology and nanoparticle synthesis have opened the floodgates for groundbreaking strategies in the production of new antimicrobial agents. A multitude of processes that vary in traits, including size, morphology, electrical charge, and surface coatings, allow researchers to develop novel composite antimicrobial substances for different applications used to perform antimicrobial activities. The antimicrobial activity of inorganic nanoparticles and carbon-based nanoparticles can be applied to various research, medical, and industrial uses in the future and offer a solution to the crisis of antimicrobial resistance to traditional approaches. In addition, nanomaterials provide a wide range of opportunities for infection prevention, diagnosis, treatment, and biofilm control. However, a detailed evaluation of these nanomaterials is required to identify their effects on natural organic tissues and to assess their impact on humans and the environment before large-scale industry implementations are carried out.

## Figures and Tables

**Figure 1 antibiotics-10-01473-f001:**
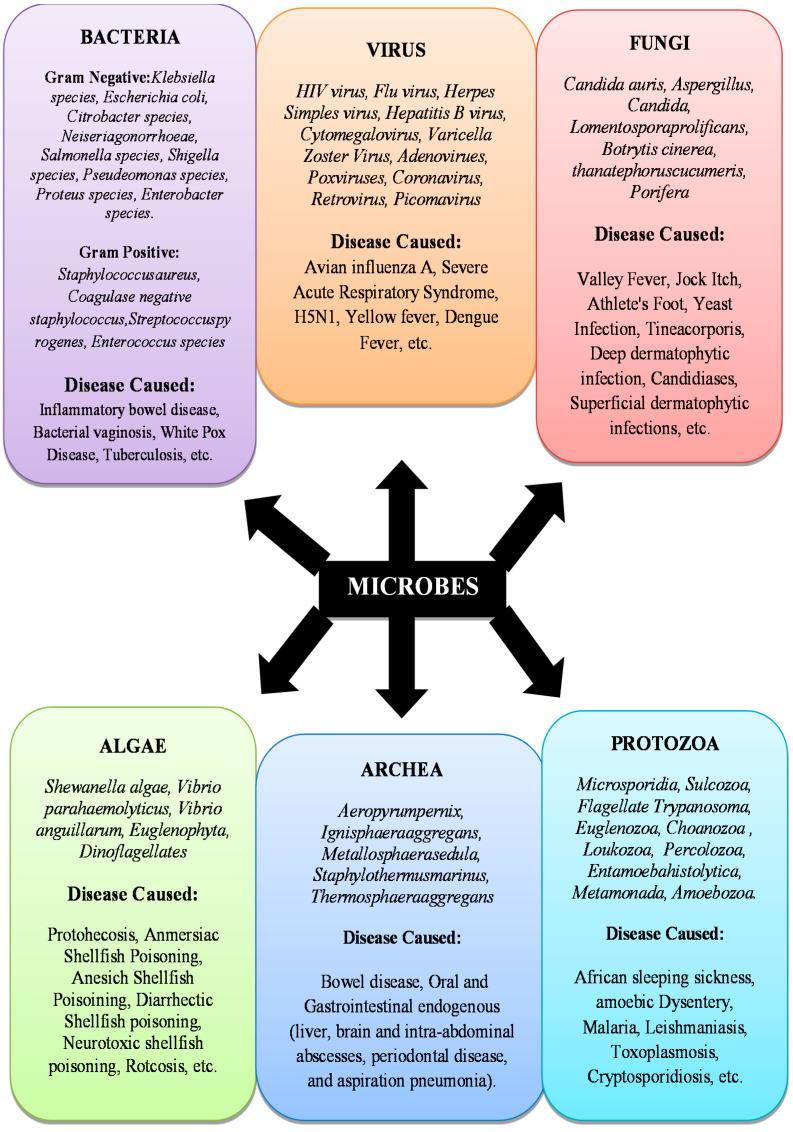
Infectious microbial species.

**Figure 2 antibiotics-10-01473-f002:**
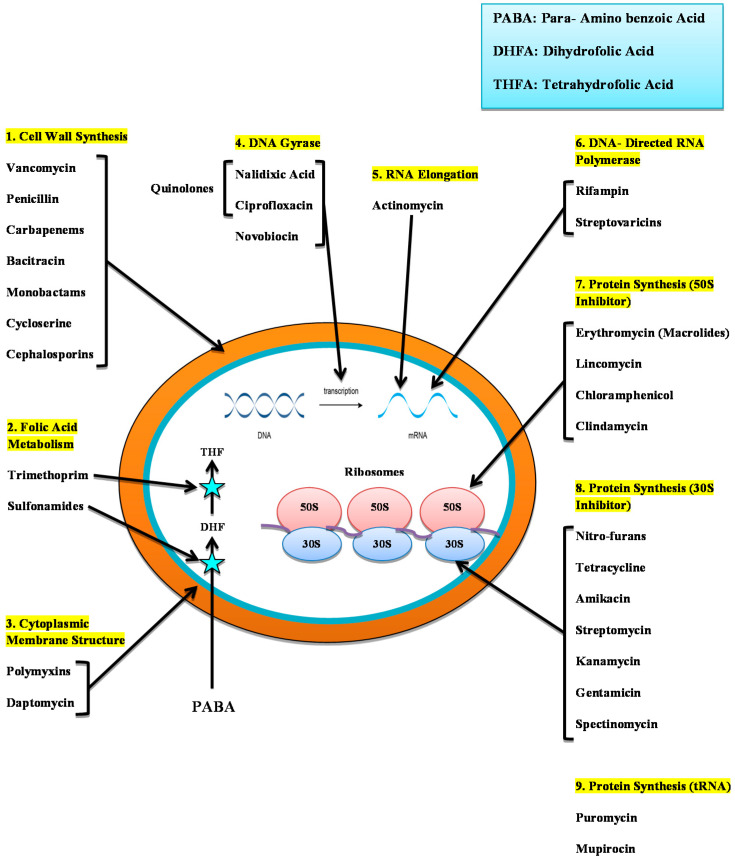
Diagrammatic representation of the mode of action of antibiotics. 1. Cell wall synthesis Inhibition. 2. Folic acid metabolism Inhibition. 3. Disruption of Cell Membranes. 4. DNA Gyrase. 5. Inhibition of RNA elongation. 6. RNA synthesis inhibitors. 7. Protein Synthesis Inhibitors (50S inhibitor). 8. Protein Synthesis Inhibitors (30S Inhibitor). 9. Inhibition of Protein Synthesis (tRNA).

**Figure 3 antibiotics-10-01473-f003:**
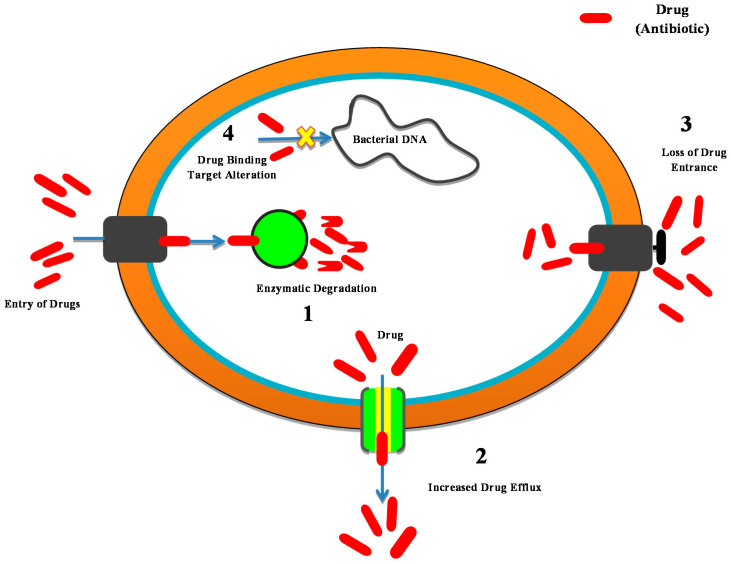
Development of the mechanism of resistance in bacteria. 1. Antibiotic enzyme inactivation/degradation. 2. Excretion of the drug through the use of efflux pumps. 3. Reduced absorption by variations in the external membrane permeability. 4. Drug target modifications.

**Figure 4 antibiotics-10-01473-f004:**
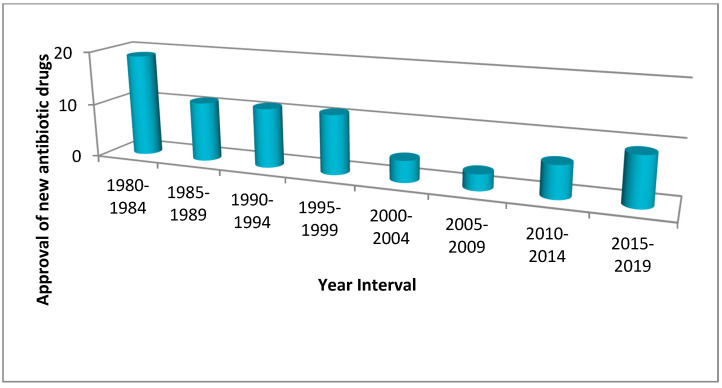
The number of approvals of new antibiotic drugs relative to year interval.

**Figure 5 antibiotics-10-01473-f005:**
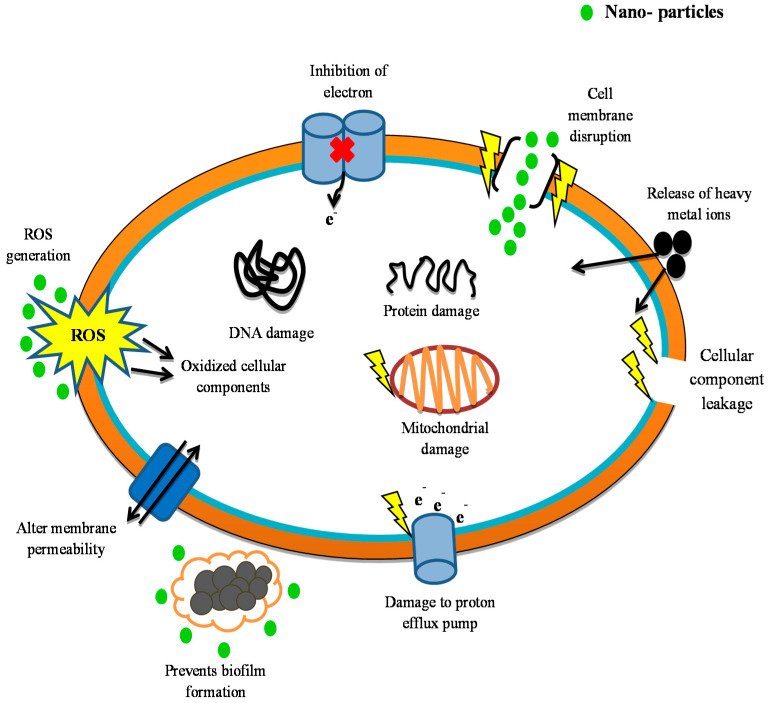
Different pathways for nanoparticle antimicrobial behavior (organic, inorganic, carbon-based, composite-based nanomaterials). ROS: Reactive Oxygen Species.

**Figure 6 antibiotics-10-01473-f006:**
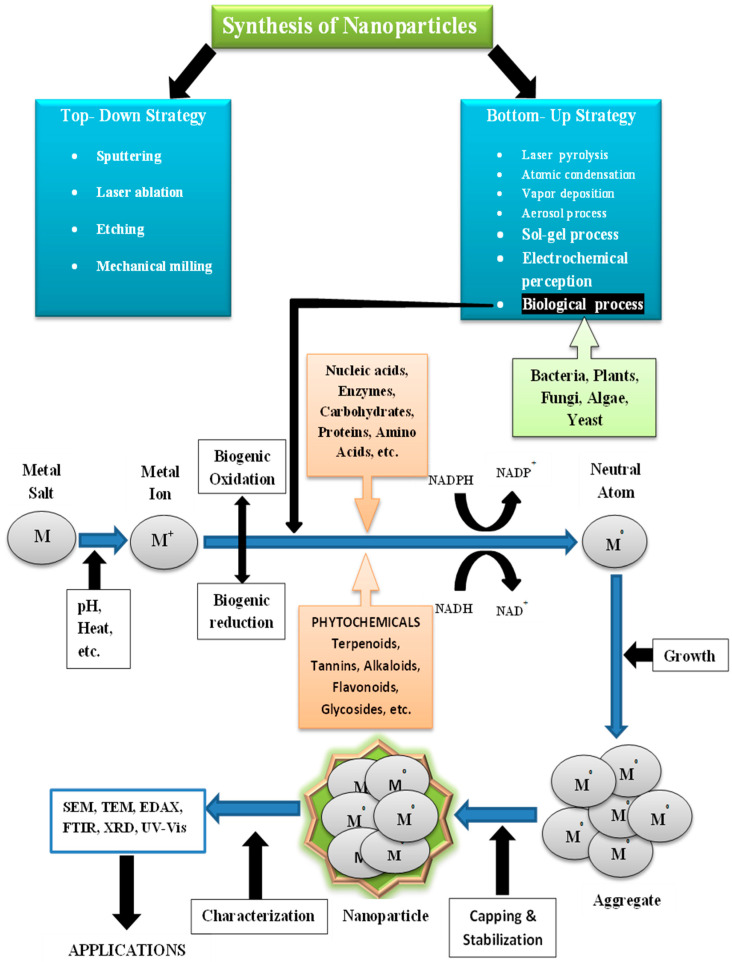
General flow chart of different procedures of nanoparticles synthesis.

**Figure 7 antibiotics-10-01473-f007:**
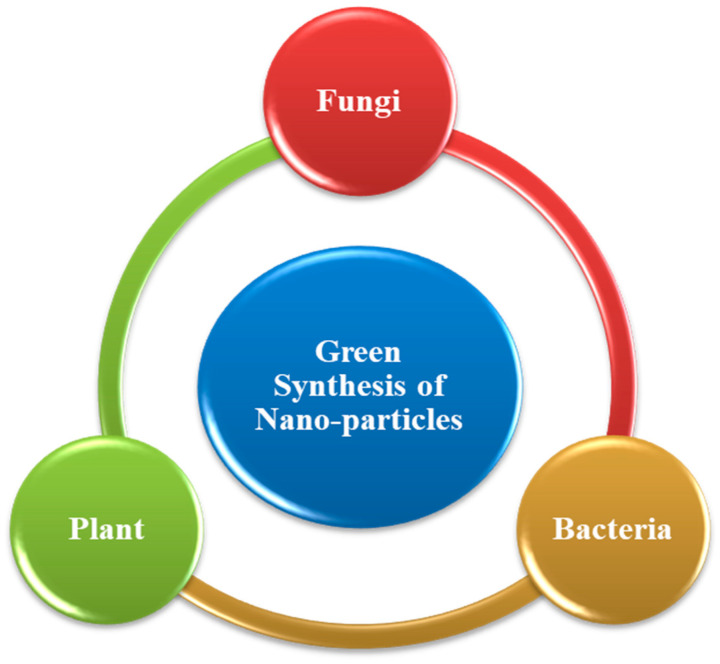
Biological synthesis of Nanoparticles.

**Figure 8 antibiotics-10-01473-f008:**
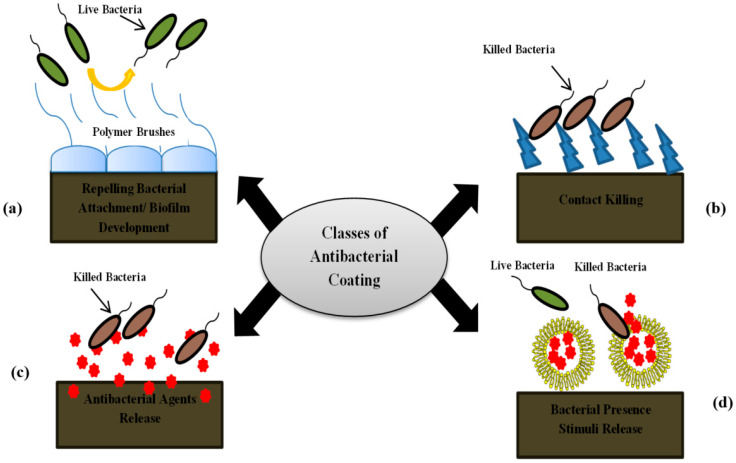
Types of antibacterial coatings: (**a**) the repelling of bacterial contact or formation of biofilms; (**b**) Contact killing; (**c**) release of antibacterial agents; (**d**) triggers the secretion of receptive compounds in the presence of bacteria.

**Figure 9 antibiotics-10-01473-f009:**
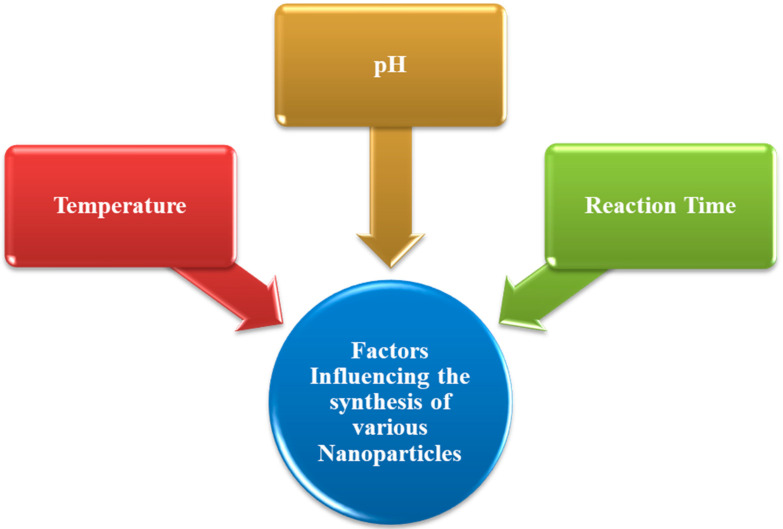
Factors influencing the synthesis of the Nanoparticles.

**Figure 10 antibiotics-10-01473-f010:**
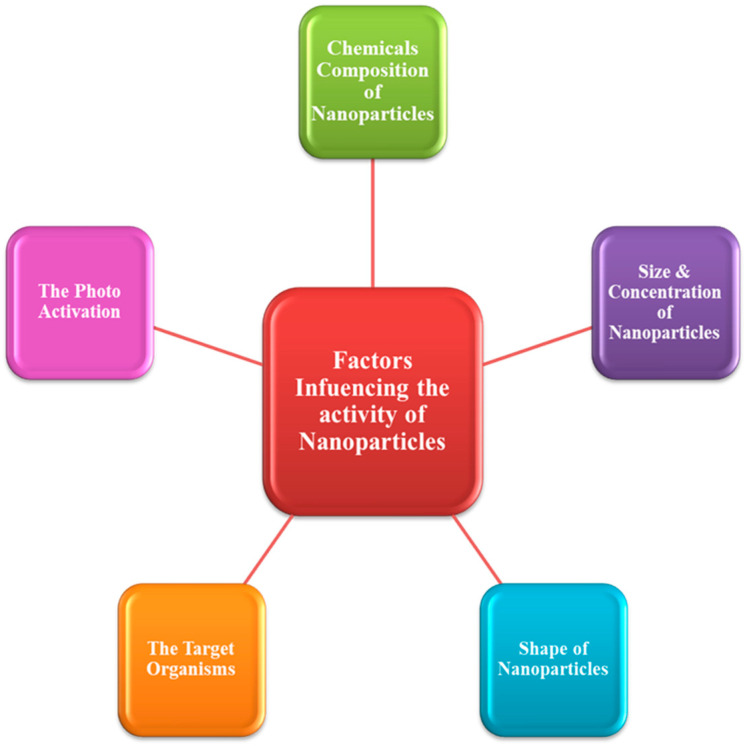
Factors influencing the activity of Nanoparticles.

**Figure 11 antibiotics-10-01473-f011:**
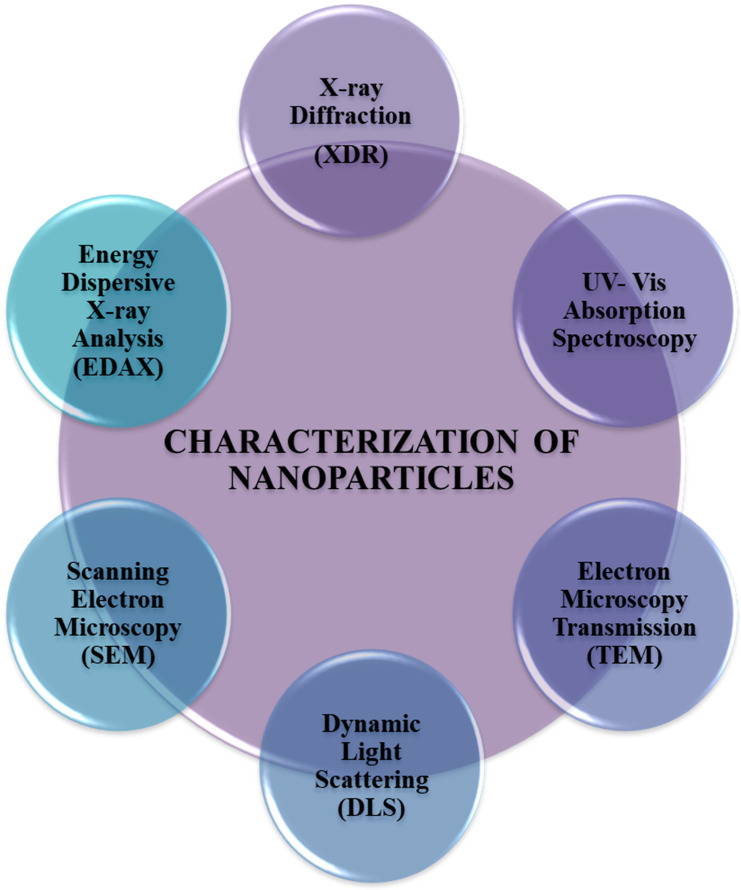
Methods for the characterization of Nanoparticles.

**Table 1 antibiotics-10-01473-t001:** History of antibiotics, their discoveries, and events that have occurred.

Time Period	Discoveries and Events	References
<19th century	Some of the oldest cultures used complex molds and plant extracts for therapy. For example, the ancient Egyptians added moldy bread to infectious wounds.	[28]
19th century	Scientists began to study the activity of antibacterial chemicals.	[29]
20th century	The most significant case in the history of antibiotics is Alexander Fleming’s discovery of penicillin in 1928. The first antibiotics were introduced in the late 1930s. The time between the 1950s and 1970s is considered the golden age in the development of new types of antibiotics, with no new classes found since then. Between 1944 and 1972, human life expectancy leaps by eight years, primarily due to the advent of antibiotics. Modified forms of erythromycin were used in the 1970s were 1980. In the 1970s, with the staled research and no discovery of new antibiotics, the fight against emerging microbial resistance to antibiotics consisted mainly in the alteration to existing antibiotics. By the 1980s and 1990s, scientists were only able to make advances in the laboratory.	[30]
21st century	Currently, more than 100 antibiotics are available to treat diseases in humans and livestock.	[31]
350 CE–550 CE	Traces of tetracycline from ancient Sudanese Nubia are found in human skeletal remains.	[32]
1887	Soil bacteria application of the Anthrax strain.	[33]
1887	Fever of intestinal Cholera infection.	[34]
1888	A bacterium substance that has antibacterial properties.	[35]
1896	Antibiotic effects found in Penicillium.	[36]
1907	A synthesis of antibiotics derived from arsenic.	[37]
1909	Arsphenamine Antisyphilitic.	[38]
1912	A Chemotherapeutic Polymer, Neosalvarsan.	[39]
1928	Synthesis of penicillin by the bacteria Staphylococcus.	[40]
1930	Decomposition of part of bacillium from soil microorganisms.	[41]
1932	Prontosil, the first microbial to receive sulphonamide.	[42]
1936	Sulfonamide	[43]
1937	Sulfonamides are added as effective antimicrobials.	[44]
1938	Sulfapyridine for the treatment of pneumococcal pneumonia is used in a clinical setting.	[45]
1939	Isolation of Tyrothricin (an antibacterial material).	[46]
1939	Gramicidin A is discovered as the first clinically effective topical antibiotic from the soil bacterium bacillus brevis.	[47]
1939	The penicillin G form became popular, the first penicillin used in therapy.	[48]
1939	Antibiotic sulfacetamide sulfonamide is first reported in the treatment of eye diseases.	[49]
1940	Sulfonamide antibiotic sulfamethoxazole is used as a common agent for the treatment of UTI and is commercialized.	[50]
1941	β-Lactam antibiotics are incorporated into clinical trials for the first time.	[51]
1941	Penicillin for therapeutic use is introduced.	[52]
1942	For the prevention of bacterial infections, sulfadimidine is used.	[53]
1942	Bacteria immune to penicillin were observed for the first time, around one year after penicillin introduction.	[54]
1942	The first antibiotic peptide was isolated from Gramicidin S.	[55]
1943	The first aminoglycoside is discovered—Streptomycin antibiotic. It is the first successful antibiotic against tuberculosis.	[56]
1943	It synthesized a drug called sulfamerazine.	[57]
1943	Penicillin was mass manufactured and used extensively during World War II to treat the Allied forces fighting in Europe.	[58]
1943	First isolates of Bacitracin. This medication is being used to treat minor cuts, burns, and scrapes causing slight skin disease.	[59]
1945	Chloramphenicol is isolated from *Streptomyces venezuelae* soil organism.	[60]
1947	Chloramphenicol is first synthesized from *Streptomyces venezuelae*, a soil organism. It was marketed in 1949, owing to its wide range of antimicrobial activity, its use subsequently becoming widespread.	[61]
1947	Chlortetracycline isolated from a sample of mud on the Missouri River. It is the first tetracycline used.	[62]
1947	The antibiotic class of polymyxin is found, the first being polymyxin B isolated from bacterium *paenibacillus polymyxa.*	[63]
1947	Nitrofuran is used in the drug class. Nitrofurans are organic, antimicrobial agents with a wide variety of activates, which include *Giardia and Salmonella spp, amebas*, *trichomonads*, and certain *coccidia*, against Gram-positive bacteria as well as Gram-negative bacteria.	[64]
1948	Mafenide isolation—an antibiotic of the form of sulfonamide, is approved by the United States FDA.	[65]
1949	The aminoglycoside antibiotic Neomycin is being isolated and used in a variety of topical product lines, such as ointments, eye drops, and creams.	[66]
1950	Oxytetracycline enters commercial use.	[67]
1950	Resistance is observed against chloramphenicol.	[68]
1952	Lincosamides, a small group of agents with a novel structure, unlike any other antibiotic, are introduced.	[69]
1952	Antibiotic thiamphenicol with a wide range of action is synthesized.	[70]
1952	Erythromycin is introduced; an antibiotic for treating bacterial inflammatory diseases, including skin infections, chlamydia infections, respiratory tract infections, syphilis, and pelvic inflammatory diseases.	[71]
1952	We add Streptogramins. Streptogramins are involved in treating vancomycin-resistant *Staphylococcus aureus* and vancomycin-resistant *Enterococcus.*	[72]
1953	Antibiotic cephalosporin C, from which cephalosporins later grow, is discovered. It prevents cell wall replication by preventing the cross-linkage of peptidoglycan.	[73]
1953	Resistance to a macrolide is observed.	[74]
1954	Benzathine penicillin is a drug for the syphilis cure.	[75]
1954	A cycloserine antibiotic is found. This is used to treat tuberculosis.	[76]
1955	First launched to the French market is the macrolide antibiotic spiramycin. Spiramycin is used for treating multiple diseases.	[77]
1956	Second, vancomycin was isolated from the orienlalis bacterium streptomyces. Vancomycin is used to treat severe joint infections, endocarditis, bloodstream infections, bone and skin infections, and meningitis caused by *staphylococcus aureus,* which is methicillin immune.	[78]
1956	Resistance is observed against erythromycin.	[79]
1957	Kanamycin is being used. It is used for the treatment of serious bacterial infections and tuberculosis.	[80]
1957	We add Ansamycins. These secondary bacterial metabolites demonstrate antimicrobial activity.	[81]
1959	Colistin becomes essential to cure Gram-negative bacterial infections.	[82]
1959	They add nitroimidazole. They are effective bactericidal agents against protozoa and anaerobes.	[83]
1960	Scientists grow methicillin to kill penicillin-resistant strains.	[84]
1960	Metronidazole is used as an important antitrichomonal agent commercially.	[85]
1961	Resistance to Methicillin is first identified.	[86]
1961	It is formulated with antibiotic ampicillin. It will become the medication of choice for the treatment of Hemophilus influenzae meningitis in a short period.	[87]
1961	It is first reported to be spectinomycin. It is used only to cure gonorrhea infections.	[88]
1961	Ethambutol is observed. The medicine is used mainly to treat tuberculosis.	[89]
1962	Fusidic acid is being incorporated into medical practice. Skin infections caused by staphylococcal bacteria are treated with antibiotics.	[90]
1962	Quinolones were mistakenly found as a by-product of studies on the chloroquine antimalarial medication.	[91]
1963	Found gentamicin. It is used to cure different kinds of bacterial infections.	[92]
1963	Gram-negative bacterium Acinetobacter baumannii becomes a pathogen and is immune to antibiotics.	[93]
1965	It is synthesized with Antibiotic Cloxacillin. It is currently effective in the treatment of a variety of bacterial infections, which include septic arthritis, cellulite, measles, external otitis, and impetigo.	[94]
1966	Resistance to Nalidixic Acid is found.	[95]
1966	Doxycycline antibiotics are synthesized. It is currently used to treat tuberculosis, bacterial pneumonia, early Lyme disease, chlamydia, syphilis, and cholera.	[96]
1966	Resistance is observed against cephalothin.	[97]
1967	First is developed clindamycin. It is commonly used for treating a variety of infections caused by bacteria.	[98]
1968	Antibiotic rifampicin is used for medical practice.	[99]
1968	Resistance to Tetracycline is found.	[100]
1968	Introduced Trimethoprim. It is primarily used for urinary infection management.	[101]
1969	Fosfomycin was found. It has a broad spectrum of action towards a vast number of Gram-negative and Gram-positive bacteria.	[102]
1970	Non-toxic, semi-synthetic acid-resistant flucloxacillin isoxazolyl penicillin is incorporated into clinical practice.	[103]
1971	Tobramycin is a discovered aminoglycoside antibiotic. It is used to treat different forms of bacterial infections, especially Gram-negative infections.	[104]
1971	Mupirocin is isolated from Pseudomonas fluorescens.	[105]
1972	The beta-lactam antibiotic cephamycin C is first isolated from the broad extracellular spectrum.	[106]
1972	Minocycline is discovered as an antibiotic. It has antibacterial and anti-inflammatory effects. Minocycline is used in acne treatment and against several infectious diseases.	[107]
1972	Tinidazole is introduced. It is an antiparasitic drug used against infections of protozoa.	[108]
1973	Carbenicillin is discovered as a bactericidal antibiotic. Carbenicillin is resistant to bactericidal action and beta-lactamase.	[109]
1974	It is a commercially available antibiotic trimethoprim/sulfamethoxazole.	[110]
1974	Cotrimoxazole is introduced. It is used in the treatment of many bacterial infections, including bronchitis, pneumonia, intestine infections, urinary tract, and skin	[111]
1976	The discovery of antibiotic amikacin. Amikacin has a broad spectrum towards a wide range of Gram-negative species, including pseudomonas, *Escherichia coli,* and certain Gram-positive species, such as *Staphylococcus aureus.*	[112]
1978	Cefoxitin comes in as an early cephamycin.	[113]
1978	The glycopeptide class of teicoplanin is discovered. Teicoplanin is used in the prophylaxis and treatment of severe Gram-positive bacterial infections, including *Enterococcus faecalis* and methicillin-resistant *Staphylococcus aureus.*	[114]
1979	Patent on cefaclor antibiotics. It is used to treat such diseases of the bacteria, for example, pneumonia and eye, lung, ear, urinary tract, and throat infections.	[115]
1981	Resistance to beta-lactamase is found at AmpC.	[116]
1981	The first fluoroquinolone, the ciprofloxacin, is discovered.	[117]
1983	Resistance is found to extended-spectrum-beta-lactamase.	[118]
1985	Discovery of daptomycin, an antibiotic.	[119]
1985	Carbapenems are introduced. They are widely used to treat severe bacterial or high-risk infections.	[120]
1986	An enterococcus immune to vancomycin is identified.	[121]
1987	It is used to treat endocarditis, intra-abdominal infections, sepsis, pneumonia, joint infections, and UTI.	[122]
1987	Extremely powerful fluoroquinolones are introduced. They are used to treat various disorders, such as the urinary tract and respiratory infections.	[123]
1987	Resistance is observed against cephalosporins.	[124]
1987	Resistance is observed against carbapenems.	[125]
1990	Resistance to fluoroquinolone is found.	[126]
1993	It is used to treat bacterial infections such as bronchitis, diarrhea, sexually transmitted diseases (STDs), and ear, lung, sinus, nose, mouth, and reproductive organs infections.	[127]
1993	Antibiotic clarithromycin is introduced. It is used in the prevention and treatment of certain bacterial infections.	[128]
1994	Cefepime moves into clinical practice. It is licensed to treat mild to severe infections.	[129]
1997	Staphyloccocus is reported to be immune to vancomycin.	[130]
1999	The quinupristin/dalfopristin antibiotic is approved.	[131]
2000	It uses oxazolidinones. These synthetic drugs are active towards a wide variety of Gram-positive bacteria.	[132]
2000	For treating infections caused by Gram-positive bacterial resistance to other antibiotics, antibiotic linezolid is introduced.	[133]
2001	In the European Union, antibiotic telithromycin is approved.	[134]
2001	Broader-spectrum fluoroquinolones are introduced.	[135]
2002	Resistance is observed against linezolid.	[136]
2002	FDA accepts cefditoren, ertapenem and pivoxil.	[137]
2002	Staphylococcus aureus is confirmed to be vancomycin-resistant.	[138]
2003	Introduce lipopeptides as antibiotics.	[139]
2003	Daptomycin is used by Gram-positive species to combat chronic and life-threatening infections.	[140]
2004	Telithromycin is introduced [60]. Certain cases of pneumonia are treated with this drug.	[141]
2005	Antibiotic tigecycline is used for the prevention of intraabdominal infections and skin and skin system infections.	[142]
2011	FDA recommends fidaxomicin to treat Difficile Infection in clostridium.	[143]
2012	FDA recommends bedaquiline for multidrug-resistant tuberculosis therapy.	[144]
2013	FDA recommends telavancin for the prevention of pneumonia in hospitals caused by susceptible Staphylococcus aureus.	[145]
2013	Centers for Disease Control and Prevention identified 17 antibiotic-resistant micro-organisms in the United States, which caused at least 23,000 deaths.	[146]
2015	The American fast-food company McDonald’s announces it will phase out its antibiotic-containing meat products.	[147]
2016	In the United States, ceftazidime/avibactam was approved for use.	[148]
2016	Natural antibiotic teixobactin is present in an uncultivated bacterial screen. Without detectable resistance, it is found to kill pathogens.	[149]
2017	Scientists develop new, safe, and simpler formulations of teixobactin-a next-generation antibiotic that beats multidrug-resistant infections such as methicillin-resistant Staphylococcus aureus.	[150]
2018	The antibiotics called odilorhabdins, or ODLs, are produced by symbiotic bacteria living in nematode worms that colonize food insects in the soil.	[151]
2019	A new family was synthesized using the so-called peptidomimetics.	[152]
2020	The newly-found corbomycin and the lesser-known complestatin have an unparalleled method of destroying bacteria which is accomplished by blocking the bacterial cell wall structure.	[153]
2021	Tebipenem hydrobromide is an oral carbapenem antibiotic in development for the treatment of complicated urinary tract infections (cUTI), including pyelonephritis, caused by susceptible microorganisms.	[154]
2021	Cefiderocol: A new cephalosporin stratagem against multidrug-resistant Gram-negative bacteria for treating complicated urinary tract infections and nosocomial pneumonia based on clinical trials demonstrating noninferiority to comparators.	[155]

**Table 2 antibiotics-10-01473-t002:** Multidrug-Resistant (MDR) species.

Organism (Species)	Resistance to Drugs	Reference
Streptococcus Pneumoniae	Multiple drugs	[169]
Streptococcus pyogenes	Tetracyclines, macrolides	[170]
Mycobacterium tuberculosis	Multiple drugs	[171]
Escherichia coli	Multiple drugs	[172]
Salmonella typhimurium	Multiple drugs	[173]
Neisseria gonorrhoeae	Penicillin, tetracycline, fluoroquinolones	[174]
Gonococci	Quinolone	[175]
Enterobacteriaceae	β-lactam (carbapenem), Quinolone	[176]
Pseudomonas aeruginosa	Multiple drugs	[177]
Enterococcus	Vancomycin	[178]
Staphylococcus aureus	β-lactam (methicillin), Vancomycin	[179]

**Table 3 antibiotics-10-01473-t003:** FDA (Food and Drug and Administration)-approved antibiotics for the treatment of Microbial infections [188].

Antibiotic Approved (FDA)	Identified Resistant Microbes	Released Year	Identification Year of Microbial Resistance
Penicillin	Penicillin-resistant *Staphylococcus aureus* Penicillin-resistant *Streptococcus pneumoniae* Penicillinase-producing *Neisseria gonorrhoeae*	1941	1942 1967 1976
Vancomycin	Plasmid-mediated vancomycin-resistant *Enterococcus faecium* Vancomycin-resistant *Staphylococcus aureus*	1958	1988 2002
Amphotericin B	Amphotericin B-resistant *Candida auris*	1959	2016
Methicillin	Methicillin-resistant *Staphylococcus aureus*	1960	1960
Extended-spectrum cephalosporins	Extended-spectrum beta-lactamase-producing *Escherichia coli*	1980	1983
Azithromycin	Azithromycin-resistant *Neisseria gonorrhoeae*	1980	2011
Imipenem	*Klebsiellapneumoniae* carbapenemase (KPC)-producing *Klebsiellapneumonia*	1985	1996
Ciprofloxacin	Ciprofloxacin-resistant *Neisseria gonorrhoeae*	1987	2007
Fluconazole	Fluconazole-resistant *Candida*	1990	1988
Caspofungin	Caspofungin-resistant *Candida*	2001	2004
Daptomycin	Daptomycin-resistant methicillin-resistant *Staphylococcus aureus*	2003	2004
Ceftazidime-avibactam	Ceftazidime-avibactam-resistant KPC-producing *Klebsiella pneumoniae*	2015	2015

**Table 5 antibiotics-10-01473-t005:** Fungal and Yeast species in the green synthesis of nanoparticles.

Fungus and Yeast	Shape of NP	Type of NP	NP Size Range (nm)	Antimicrobial Effect of NP	References
➢ *MKY3*	Hexagonal	Ag	2–5	*Against S. aureus*	[389]
➢ *Volvariellavolvacea*	Hexagonal/Spherical	Au and Ag	20–150	*Antibacterial*	[390]
➢ *Aspergillusflavus*	Oval	TiO_2_	60–74	*Against* *E. coli and S. aureus*	[391]
➢ *Streptomyces sp. NH21*	Spherical	Au	10	*Against E. coli, K. pneumoniae, P. mirabilisS. infantis, P. aeruginosa and* *B. subtilis*	[392]
➢ *Alternaria sp*	Spherical	Ag	80	*Against B. subtilis, S. aureus, E. coli and S. marcescens,*	[393]
➢ *Penicillium*	Spherical	Ag	10–100	*Against B. cereus, S. aureus, E. coli and P. aeruginosa*	[394]
➢ *A. terreus*	Spherical	Au	10–19	*Escherichia coli*	[395]

**Table 6 antibiotics-10-01473-t006:** Bacterial species in the green synthesis of nanoparticles.

Bacteria	Shape of NP	Type of NP	NP Size Range (nm)	Antimicrobial Effect of NP	References
➢ *Escherichia coli*	Spherical	CdS	2–5	*Against* *E. coli strain BW25113*	[398]
➢ *Strains NS2 and NS6*	Crystal structures	PbS	40–70	*Bioremediation*	[399]
➢ *Bacillus mycoides*	Spherical	TiO_2_	40–60	*Suppress aquatic biofilm growth*	[400]
➢ *Aeromonashydrophila*	Spherical	ZnO	50–70	*Against* *P. aeruginosa and A. flavus*	[401]
➢ *Proteus mirabilis PTCC1710*	Spherical	Au	10–20	*No data available*	[402]
➢ *E. faecalis*	Spherical	ZnO	16–96	*Against S. aureus, K. pneumonia, and E. Coli*	[403]
➢ *E. faecalis*	Spherical	ZnO	16–96	*Against K. pneumonia 125,* *E. coli 03,* *E. coli MTCC 9537,* *S. aureus 20, S. flexneri MTCC 1457, K. pneumonia MTCC 109,* *P. aeruginosaMTCC 741, S. aureus MTCC 96,*	[403]

**Table 7 antibiotics-10-01473-t007:** Plant species in the green synthesis of nanoparticles.

Plant	Shape of NP	Type of NP	NP Size Range (nm)	Anti-Microbial Effect of NP	References
➢ *Camellia Sinensis*	Spherical/Triangular	ZnO	30–40	Antibacterial	[408]
➢ *Catharanthusroseus*	No typical shape	TiO_2_	25–110	No data available	[409]
➢ *Geranium leaves*	Quasilinear	Ag	40	Antimicrobial	[410]
➢ *Avena sativa*	Rod-shaped	Au	5–20	No data available	[411]
➢ *Phyllanthusamarus*	Spherical	CuO	20	Against *B. subtilis*	[412]
➢ *E. japonica*	Spherical	Au	46–70	Against *E. coli and S. aureus*	[413]
➢ *J. adhatoda L*	Spherical	Au	5–50	Against *P. aeruginosa*	[414]
➢ *T. procumbens*	-	Fe	80–100	Against *P. aeruginosa*	[415]
➢ *G. jasminoides and L. inermis*	Hexagonal	Fe	21	*S. aureus, E. coli, P. mirabilis andS. enterica.*	[416]
➢ *C. cujete L*	Spherical	Au	32–89	*Salmonella typhi (MTCC 531), P. aeruginosa (MTCC 1688), E. coli (MTCC 1687), B. subtilis (MTCC 441), V. cholerae* *(MTCC 3906) and S. flexneri (MTCC 9543).*	[417]
➢ *C. pictus D. Don*	Hexagonal	ZnO	11–25	*S. paratyphi (NCIM 2501), B. subtilis (NCIM 2063), S. aureus (NCIM 2079), and E. coli* *(NCIM 2065).*	[418]

**Table 8 antibiotics-10-01473-t008:** Comparison of Nanoparticles with antibiotics and their combination.

Features	Antibiotics	Nanoparticles	Combination	Reference
Size	Complex because of the poor membrane transport and size scale	The ultra-small size is controllable and can penetrate membranes easily	The small size of the NP carriers makes it easier to transport the antibiotics	[468]
Protection	This shows resistance against bacteria, all because of the increased efflux and decreased uptake.	No resistance against bacteria and shows a strong effect on bacteria.	NP carriers can help protect the drugs from resistance by target bacteria by increasing the serum levels of antibiotic	[469]
Precision and safety	Not targeted at the specific location and thereby shows adverse effects	Helps target the specific areas and thereby minimize the adverse effects.	More specific targeting and minimal adverse effects	[470,471]
Controllability	Uncontrollable release of the drug	Controlled release of the drug	Controlled release of the drug	[472,473,474]
Bioavailability	Low bioavailability and easily biodegradable	Improved bioavailability and non-degradable	Improved bioavailability and non-degradable	[475]
Enzymatic degradation	These can be degraded enzymatically	Cannot be degraded enzymatically	Cannot be degraded enzymatically	[475]

**Table 9 antibiotics-10-01473-t009:** Nano-particles and their applications on animal models.

Animal Model	Nano-Particles	Applications	Reference
Piglets	Nano zinc	Diarrhea in young piglets can be reduced by Nano zinc	[479]
Albino Rats	Silver	AgNPs using A. nobilis revealed higher microbicidal activity for wound healing.	[480]
Mouse	Gold	In xenograft mouse models, QAuNPs significantly inhibited cell proliferation, caused apoptosis in vitro, and destroyed angiogenesis and tumor regression in vivo	[481]
Mouse	Copper	Treatment of wounds in mice with copper nanoparticles.	[482]

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
