# Peer review of "Nanotechnology as a Novel Approach in Combating Microbes Providing an Alternative to Antibiotics"

_antibiotics, 2021, doi:10.3390/antibiotics10121473_

Round 1

Reviewer 1 Report

1) The concept of “biogenic nanoparticles” has to be included. What did you mean? What are the differences between the “ordinary” nanoparticles and “biogenic nanoparticles”? Maybe the concept has to come in the introduction. The idea is present in about lines 861-881.

2) Figures 1, 6, 9, and 11 are too elementary. They could remove them.

3) Lines 135 to 139 could be included in the sentence of the previous paragraph, not like itemized sentences.

4) Table 1 is significantly big. The first page of the Table (page 5) presents an overview century by century. But, pages 6 to 11 shows details year by year. I believe such deep chronological details could be removed, or the authors could split the table into two tables: 1) concerning the overview on page 5; 2) the history year by year. Mainly, I prefer two tables; once the authors carried out one in-depth review of the history, we should not lose this panorama.

5) Concerning the lines 391 to 394:

The nanomaterials are classified into four categories (see Table 4):

  1. Carbon-based nanomaterial
  2. Inorganic nanomaterial
  3. Organic-based nanomaterial
  4. Composite-based nanomaterial

Comment: the authors proposed classifying the nanomaterials in four categories, but such categories are entirely mixed in the table. I believe that it could divide into four tables and then every table could be briefly described. Each table could come in the respective section (4.3.1, 4.3.2, 4.3.3, 4.3.4

6) Section : Quaternary Ammonium Compounds (line 649)

Comment: No reference was provided in this section (lines 650-657)

7) Pay attention to the table format! They are not following a standard model.

8) Figures 7 and 8 could be changed. I meant: Figure 7 becomes Figure 8 and Figure 8 becomes Figure 7.

4.6.2. Purification of Nanoparticles Extracted Biologically (it describes the Figure 8)

4.6.3. Nanoparticles Coating for Antibacterial Activity (it describes the Figure 7)

9) Line 1131: 7. Antimicrobials and Nan-Particles In Combination

Comment: Nanoparticles instead of Nan-Particles

Author Response

Review 1

Comments and Suggestions for Authors

  • The concept of “biogenic nanoparticles” has to be included. What did you mean? What are the differences between the “ordinary” nanoparticles and “biogenic nanoparticles”? Maybe the concept has to come in the introduction. The idea is present in about lines 861-881.

Author’s response:

We have incorporated details as suggested by reviewer:

“The biogenic nanoparticle synthesis includes green methodologies. Green synthesis advantages include; the production of stable nanoparticles, the use of a biomass-based surface coating that provides extra active surface areas for biological interaction, the exclusion of dangerous formation of byproducts and additional stabilizing or reducing factors that eventually makes the procedure economical [15,16]”.

  • Figures 1, 6, 9, and 11 are too elementary. They could remove them.

Author’s response:

Figure one has been removed while the other figures are relevant to the topic therefore can be included.

  • Lines 135 to 139 could be included in the sentence of the previous paragraph, not like itemized sentences.

Author’s response:

We changed it according to the reviewer suggestion as”

These differences are due to their mechanism of action on the body, genetic content of the microbe, the function of the microbe, the shape of the microbe and size of the microbe”.

  • Table 1 is significantly big. The first page of the Table (page 5) presents an overview century by century. But, pages 6 to 11 shows details year by year. I believe such deep chronological details could be removed, or the authors could split the table into two tables: 1) concerning the overview on page 5; 2) the history year by year. Mainly, I prefer two tables; once the authors carried out one in-depth review of the history, we should not lose this panorama.

Author’s response:

Adjusted table as suggested by the reviewer.

5) Concerning the lines 391 to 394:

The nanomaterials are classified into four categories (see Table 4):

  1. Carbon-based nanomaterial
  2. Inorganic nanomaterial
  3. Organic-based nanomaterial
  4. Composite-based nanomaterial

Comment: the authors proposed classifying the nanomaterials in four categories, but such categories are entirely mixed in the table. I believe that it could divide into four tables and then every table could be briefly described. Each table could come in the respective section (4.3.1, 4.3.2, 4.3.3, 4.3.4).

Author’s response:

Adjusted the tables as suggested by the reviewer. Table 4a to Table 4d

These nano-materials typically have carbon and are found in spheres, ellipsoids, and hollow tubes. The types of carbon-based NMs include Fullerenes (C60), Carbon Nanotubes (CNT), Carbon Nano-fibers, Carbon Black, Graphene (Gr) [223] (Table 4a.). Metal and metal oxide Nanoparticles are included in such NMs. These nanomaterials can be synthesized into metals like Ag or Au, metal oxides like ZnO and TIO2 NP, and semiconductors like ceramics and silicon. A few of these demonstrate high antimicrobial activity towards certain bacteria, viruses as well as other microbes. Specific nanomaterials with strong antibacterial efficacy are thought to have a high volume to surface ratios as well as a special chemical–physical property [237] (Table 4b.)

6) Section : Quaternary Ammonium Compounds (line 649)

Comment: No reference was provided in this section (lines 650-657)

Author’s response:

We have added reference (493) line number 671.

7) Pay attention to the table format! They are not following a standard model.

Author’s response:

We have formatted the table according to the reviewer suggestion.

8) Figures 7 and 8 could be changed. I meant: Figure 7 becomes Figure 8 and Figure 8 becomes Figure 7.

Author’s response:

Changes made as suggested.

.

4.6.2. Purification of Nanoparticles Extracted Biologically (it describes the Figure 8)

4.6.3. Nanoparticles Coating for Antibacterial Activity (it describes the Figure 7)

9) Line 1131: 7. Antimicrobials and Nan-Particles In Combination

Comment: Nanoparticles instead of Nan-Particles

Author’s response:

The figures have been placed according to their description and “Nan-Particles” is replaced with “Nanoparticles” as suggested by the reviewer (line #1234).

Reviewer 2 Report

The manuscript by Bismillah et al. entitled “Nanoparticles as a novel approach in combating microbes providing an alternative to antibiotics” is based on nanoparticles as a novel approach in combating microbial plethora and its resistance to the antibiotic. Although several papers have been published on Nanotechnology as a therapeutic tool against microbial agents, the focus of this paper is quite different from the conventional review papers. Nonetheless, I have some concerns before it can be considered for acceptance.

The title is misleading; besides, nanoparticle(s) is not an approach, but nanotechnology is an approach. In addition, many readers might get confused that nanoparticles could be a replacement for antibiotics, which is not valid. So, revise the title.

  1. The abstract does not give proper information. Abstract means a full-fledged summary that should highlight the information and topics covered in the manuscript. Also, highlight essentialities and future perspectives of the study.
  2. Line 56-61 merge all sentences to cover all the points.
  3. Line 89-95 merge all sentences to cover all the points.
  4. In Figures 1, 2, 3 References are mentioned with legends. Are these Figures copied from Article? If yes, do you have copyright permission? If it is, self-drawn figures then remove references from legends.
  5. Figure 1 is so simple and irrelevant to the topic. There is no need to teach the classification of microbes in this review. Kindly remove it.
  6. Italicize et al., in the tables and at various places in the manuscript and need to correct (Table 4) not (see table 4) and it is needed to change everywhere in the manuscript. Just write table number no need to write see table.
  7. Correct (figure 8) not (see figure 8) and same for figure 6 under heading 4.6.1. and figure 7 in 4.6.3.
  8. Cuperous oxide formula is CU2O in place of CU2O. Likewise TiO2 NPs in place of TiO2 NPs and Correct TiO2. It should be TiO2 under heading # 5.
  9. In Table 1, the Authors covered updates of antibiotics up to 2020. What about 2021?

Overall, it is a well-written review. However, the authors should improve the quality of the figures.

Author Response

The manuscript by Bismillah et al. entitled “Nanoparticles as a novel approach in combating microbes providing an alternative to antibiotics” is based on nanoparticles as a novel approach in combating microbial plethora and its resistance to the antibiotic. Although several papers have been published on Nanotechnology as a therapeutic tool against microbial agents, the focus of this paper is quite different from the conventional review papers. Nonetheless, I have some concerns before it can be considered for acceptance.

 Author’s response:

The authors are grateful to the reviewer for their valuable comments and suggestions.

The title is misleading; besides, nanoparticle(s) is not an approach, but nanotechnology is an approach. In addition, many readers might get confused that nanoparticles could be a replacement for antibiotics, which is not valid. So, revise the title.

Author’s response:

The title has been revised as suggested by the reviewer: the new title is ‘Nanotechnology as a novel approach in combating microbes providing an alternative to antibiotics’. The changes have been indicated as red font.

  1. The abstract does not give proper information. Abstract means a full-fledged summary that should highlight the information and topics covered in the manuscript. Also, highlight essentialities and future perspectives of the study.

Author’s response:

We rewrite the abstract and added all the aspects discussed in the review paper. 

Line 56-61 merge all sentences to cover all the points

Author’s response:

Alterations have been made in the text as per suggestions by the reviewer. The paragraph below with changes has been highlighted in the text as red font.

Green synthesis advantages include: the production of stable nanoparticles, the use of a biomass-based surface coating that provides extra active surface areas for biological interaction, the exclusion of dangerous formation of byproducts, additional stabilizing or reducing factors that eventually makes the procedure economical [15,16].

Line 89-95 merge all sentences to cover all the points.

Author’s response:

Alterations have been made in the text as per suggestions by the reviewer. The paragraph below with changes has been highlighted in the text as red font.

They differ in many respects, including: mechanism of action on the body, genetic content of the microbe, the function of the microbe, the shape of the microbe, and size of the microbe.

  1. In Figures 1, 2, 3 References are mentioned with legends. Are these Figures copied from Article? If yes, do you have copyright permission? If it is, self-drawn figures then remove references from legends.
  2. Author’s response:

The figures were adopted from different studies.  We have drawn the figures on the basis of knowledge available in the literature. Hence there is no need for copyrights.

  1. Figure 1 is so simple and irrelevant to the topic. There is no need to teach the classification of microbes in this review. Kindly remove it.
  2. Author’s response:

The figure 1 has been deleted and remaining figures have been re numbered in the text accordingly.

  1. Italicize et al., in the tables and at various places in the manuscript and need to correct (Table 4) not (see table 4) and it is needed to change everywhere in the manuscript. Just write table number no need to write see table.

Author’s response:

We have removed the names of authors as the numbers were already given according to journal format. We also replaced “see table” with“ table” only

  1. Correct (figure 8) not (see figure 8) and same for figure 6 under heading 4.6.1. and figure 7 in 4.6.3.

Author’s response:

We have corrected “see figure” to “figure” in the text.

  1. Cuperous oxide formula is CU2O in place of CU2O. Likewise TiO2 NPs in place of TiONPs and Correct TiO2. It should be TiOunder heading # 5.

Author’s response:

Corrected to CU2O as suggested in line# 330

In Table 1, the Authors covered updates of antibiotics up to 2020. What about 2021?

Author’s response:

We have added some relevant updates in table 1 as suggested.

Overall, it is a well-written review. However, the authors should improve the quality of the figures.

Author’s response:

We are thankful for the comments of the reviewer and we have improved the quality of the figure as suggested

Reviewer 3 Report

  1. References need to be past 6 years unless important.
  2. Manuscript Need table on its application in vivo animal model.
  3. Authors should improve quality figures and also added other figures for other section in the manuscript.
  4. The clinical applications and studies are missing.
  5. The English needs to be checked and corrected by a native English writer.

Author Response

  1. References need to be past 6 years unless important.

Author’s response:

Most of the references included in the review article are within the suggested range. However, the description of some antibiotics needed some old references. Hence, the authors had to incorporate some old references as well.

  1. Manuscript Need table on its application in vivo animal model.

Author’s response:

We have added the applications in vivo animals in lines 1271-1276 and added table 9.

  1. Authors should improve quality figures and also added other figures for other section in the manuscript.

Author’s response:

We have improved the quality of figures. There are already 10 figures and the authors think, for a review article this number of figures is sufficient.

  1. The clinical applications and studies are missing.

Author’s response:

The article has been written for a comparative analysis of classical antibiotics and nanotechnology based advanced materials for the management of microbial infections. It is really a very good suggestion to cover the clinical applications in the text. However, the authors have a plea that the addition of another aspect will be out of our aims and scope, it will further increase the length of review article. We have a plan to write a separate article on clinical applications and trials of drug coasted nanoparticles. 

  1. The English needs to be checked and corrected by a native English writer.

Author’s response:

The authors have a vast experience of manuscripts writing in English language with 100s of published research articles and reviews. However, as suggested by the reviewer, we have improved the quality of English language using online software ‘grammarly’ and ginger.  

Round 2

Reviewer 1 Report

No comments.